# MEMORY-EFFICIENT SELF-SUPERVISED CONTRASTIVE LEARNING WITH A SUPERVISED LOSS

## ABSTRACT

Contrastive Learning (CL) is among the most popular methods for self-supervised learning (SSL). However, CL requires a large memory and sample size and careful hyperparameter tuning. These factors make it difficult to learn high-quality representations with limited amount of memory. In this work, we theoretically analyze a recently proposed *supervised* approach, DIET, for SSL. DIET labels every example by its datum index and trains on the labeled data with a supervised loss. DIET does not require a large sample size or hyperparameter tuning. However, it does not scale to larger datasets due to the massive classifier head and does not always match the performance of existing methods. Given its remarkable simplicity and inconsistent results, it is not obvious whether DIET can achieve the performance of CL methods, which explicitly model pairwise interactions between augmented examples. We prove that, perhaps surprisingly, for a linear encoder DIET with MSE loss is equivalent to spectral contrastive loss. Then, we prove that DIET is prone to learning less-noisy features and may not learn all features from the training data. We show feature normalization can provably address this shortcoming and use of a projection head can further boost the performance. Finally, we address the scalability issue of DIET by reducing its memory footprint. The modified approach, namely SCALED-DIET (S-DIET), substantially improves on the linear probe accuracy of DIET across a variety of datasets and models and outperforms other SSL methods, all with limited memory and without extensive hyperparameter tuning. This makes S-DIET a promising alternative for simple, effective, and memory-efficient representation learning.

## 1 INTRODUCTION

Contrastive Learning (CL) has emerged as one of the most successful methods to learn generalizable features without the need for labels. CL trains an encoder by aligning augmented views of the same example, and pushing augmented views of different examples apart (Chen et al., 2020; Zbontar et al., 2021; Chen & He, 2021; Grill et al., 2020). However, CL has a complicated pairwise loss function that requires large memory and sample size to effectively align representations of similar examples (Huang et al., 2022), and needs careful hyperparameter tuning (Khosla et al., 2020). These factors make it difficult to learn high-quality representations with CL. This raises a key question: *are there simpler ways to learn high-quality representations with small memory?*

Recently, Balestriero (2023) proposed a *supervised* alternative for representation learning, namely DIET, which labels every example by its datum index and trains on the labeled data with Cross Entropy loss. DIET obtains state-of-the-art generalization performance when learning representations from small datasets, and does not require extensive hyperparameter tuning. However, it does not scale to larger datasets as the huge classifier head cannot be fit into the memory, and does not achieve competitive performance on all benchmarks. This raises key questions about the theoretical and practical viability of DIET as an alternative for SSL:

- Theoretically, how do the solutions learned by DIET and CL compare?
- Why might DIET fail to achieve good performance on some benchmarks?
- Can DIET be implemented in an efficient, scalable manner?

In this work, we address each of these questions. First, by studying a linear encoder we prove that, perhaps surprisingly, DIET with MSE loss is equivalent to the spectral contrastive loss, Then, we show that DIET is highly prone to learning the less-noisy and easier to learn features instead of all task-relevant features. To address this, we prove that normalizing the features before the classification head enhances the feature learning ability of DIET. We also show that the use of projection head can further boost the performance. Finally, we propose a modified loss function and parameter update step to reduce the memory requirements of DIET. In doing so, our modified DIET, namely SCALED-DIET (S-DIET), *state-of-the-art performance with limited memory* on a variety of datasets and model architectures without extensive hyperparameter tuning, providing a promising alternative for memory-efficient SSL.

We conduct extensive experiments on CIFAR-10, CIFAR-100 (Krizhevsky et al., 2009), ImageNet-100 (Deng et al., 2009), and TinyImageNet (Le & Yang, 2015), and show that S-DIET significantly improves the performance of DIET and outperforms CL and other self-supervised learning methods under limited memory requirements. We also conduct an ablation study to confirm the effectiveness of feature normalization and projection head.

## 2 RELATED WORK

**Self-Supervised Contrastive Learning.** Self supervised learning (SSL) methods broadly aim to learn representations that capture semantically meaningful features of the data. Several works such as SimCLR (Chen et al., 2020) and MoCo (He et al., 2020) demonstrated the effectiveness of the contrastive or InfoNCE loss (Oord et al., 2018), which aims to maximize the similarity of so-called positive pairs, while minimizing the similarity of all other pairs to avoid representation collapse. Since then, the general framework of designing pairwise losses that compare different views of the data has proven a popular and effective approach in SSL. BYOL (Grill et al., 2020) showed that the use of negative pairs in the contrastive loss is unnecessary, instead using an online and target network to avoid collapse. SimSiam (Chen & He, 2021) developed a method based on siamese networks which also does not require negative pairs. Barlow Twins (Zbontar et al., 2021) proposed a new loss function which includes a redundancy reduction term to avoid representational collapse. The introduction of more advanced data augmentations (Peng et al., 2022; Yang et al., 2022) and new methods for selecting positive pairs (Dwibedi et al., 2021) have also boosted performance. But these SSL methods have complicated loss functions which often require maintaining multiple views of the same example, large batch sizes, and careful hyperparameter tuning. These factors increase memory requirements and make it difficult to apply to new tasks.

**DIET.** Recently, Balestriero (2023) proposed a *supervised* alternative for representation learning, namely DIET, which assigns labels to every example by its datum index and trains on the labeled data with Cross Entropy Loss. DIET does not require a large sample size or careful data augmentation or hyperparameter tuning. However, it falls short on some benchmarks and is memory intensive due to the massive classifier head, a fatal limitation when scaling to larger datasets. In our work, we will address these shortcomings.

**Theory on Contrastive Learning.** There has been much progress on theoretically understanding CL. Wang & Isola (2020); Graf et al. (2021) study the clustering structure of learned embeddings. Arora et al. (2019); HaoChen et al. (2021); Lee et al. (2021); Tosh et al. (2021) provide provable guarantees for downstream task performance. Wen & Li (2021); Ji et al. (2021) analyze the feature learning power of contrastive learning. Saunshi et al. (2022); HaoChen & Ma (2022); Xue et al. (2023) analyze the role of inductive biases in the successes and failures of CL. Xue et al. (2024) investigates the benefits of using a projection head, a common technique in CL. Other works relate CL to different methods such as generalized multi-dimensional

scaling (Balestriero, 2023), a conditional energy based model (Murphy, 2022), or kernel learning (Johnson et al., 2023). But these works do not provide a precise comparison between the representations learned by supervised and contrastive learning methods. Our work provides the first rigorous theoretical connection between CL and supervised learning by defining a precise correspondence between global minima of a supervised and a contrastive loss.

## 3 BACKGROUND: CONTRASTIVE LEARNING AND DIET

In the SSL setting, we are given a set of input examples $\{\boldsymbol{x}_i\}_{i=1}^n \subset \mathbb{R}^d$ without labels, and the goal is to construct an embedding map $f : \mathbb{R}^d \to \mathbb{R}^m$ such that the embeddings capture the semantically meaningful features of the data. A simple yet effective starting point is to assign each example a distinct label, and then obtain multiple examples per class by performing a set of augmentations $\mathcal{A}$ on the original example, thereby constructing the labeled dataset

$$\mathcal{D} = \{(A(\boldsymbol{x}_i), i) : A \in \mathcal{A}, i = 1, \ldots, n\}. \tag{1}$$

A pair of examples with the same label, namely two examples that are augmentations of the same original input, is known as a positive pair, while all other pairs are known as negative pairs.

**Contrastive Learning (CL).** A popular and effective loss function known as the contrastive or InfoNCE loss aims to maximize the cosine similarity of positive pairs while minimizing the cosine similarity of negative pairs (Oord et al., 2018; Chen et al., 2020). Formally, letting $\mathcal{P}_{pos}$ be the distribution over positive pairs and $b$ be the batch size, we define

$$\mathcal{L}_{cl} = - \underset{(\boldsymbol{x}_1, \boldsymbol{x}_2) \sim \mathcal{P}_{pos}}{\mathbb{E}} \left[ \frac{\text{sim}(f(\boldsymbol{x}_1), f(\boldsymbol{x}_2))}{\tau} \right] + \underset{\boldsymbol{x}_1, \ldots, \boldsymbol{x}_b \sim \mathcal{D}}{\mathbb{E}} \left[ \log \left( \sum_{i=1}^b \exp \left( \frac{\text{sim}(f(\boldsymbol{x}), f(\boldsymbol{x}_i))}{\tau} \right) \right) \right].$$

Here $\text{sim}(\cdot, \cdot)$ is the cosine similarity and $\tau$ is the temperature of the softmax distribution.

A variant of this loss function known as the spectral contrastive loss has also proven popular in theoretical analysis (HaoChen et al., 2021; HaoChen & Ma, 2022; Saunshi et al., 2022; Xue et al., 2023). It takes the form

$$\mathcal{L}_{scl} = - \underset{(\boldsymbol{x}_1, y_1), (\boldsymbol{x}_2, y_2) \sim \mathcal{D}}{\mathbb{E}} \left[ \delta_{y_1, y_2} f(\boldsymbol{x}_1)^\top f(\boldsymbol{x}_2) \right] + \frac{1}{2} \underset{(\boldsymbol{x}_1, y_1), (\boldsymbol{x}_2, y_2) \sim \mathcal{D}}{\mathbb{E}} \left[ (f(\boldsymbol{x}_1)^\top f(\boldsymbol{x}_2))^2 \right], \tag{2}$$

where $\delta$ is the Kronecker delta. [1]

**Limitations.** CL and its variants have proven remarkably successful. However, due to its complicated pairwise loss function, CL requires a large sample size $n$, a large-capacity encoder $f$, and carefully tuning hyperparameters, such as $\tau$ and $b$. In addition, the necessity to maintain multiple views of the same example and use large batch sizes $b$ increases memory requirements.

**DIET.** In contrast, DIET presents an alternative approach (Balestriero, 2023). Moving away from pairwise losses, DIET instead appends a linear classifier $\boldsymbol{W}_H \in \mathbb{R}^{n \times m}$ and applies a supervised loss $l$ to $\boldsymbol{W}_H f$, i.e.,

$$\mathcal{L}_{diet}^l = \mathbb{E}_{(\boldsymbol{x}, y) \sim \mathcal{D}}[l\left((\boldsymbol{W}_H f)(\boldsymbol{x}), y\right)]. \tag{3}$$

In practice, $l$ is the cross entropy loss with label smoothing, although we will also consider other losses such as mean-squared error in our analysis.

**Limitations.** DIET does not require a large sample size or hyperparameter tuning. However, it does not scale to larger datasets due to the massive classifier head $\boldsymbol{W}_H$, which grows with the number of examples in the dataset, and does not match the performance of state-of-the-art SSL methods across all benchmarks.

---

[1] We remark that Eq. 2 differs from some previous definitions by a few constant factors. This does not affect any of the analysis, see Appendix A.2 for further discussion.

## 4 UNDERSTANDING REPRESENTATION LEARNING WITH DIET

While DIET and contrastive learning appear to be unrelated at first glance, we prove an unexpected equivalence in the case that $f$ is a linear encoder. Specifically, we compare the global minimizers of the spectral contrastive loss with the global minimizers of DIET with mean squared error loss and one-hot encoded labels:

$$l((\boldsymbol{W}_H f)(\boldsymbol{x}), y) = MSE((\boldsymbol{W}_H f)(\boldsymbol{x}), \boldsymbol{e}_y) = \frac{1}{2}\|(\boldsymbol{W}_H f)(\boldsymbol{x}) - \boldsymbol{e}_y\|^2,$$

$$\mathcal{L}_{diet}^{mse} = \mathbb{E}_{(\boldsymbol{x},y)\sim\mathcal{D}}[MSE((\boldsymbol{W}_H f)(\boldsymbol{x}), y)].$$

A priori, it cannot be expected that minimizing $\mathcal{L}_{diet}^{mse}$ induces any particular structure on the embeddings $\{f(\boldsymbol{x}_i)\}$ since the classifier head $\boldsymbol{W}_H$ can perform an arbitrary linear transformation on the embeddings. To make a meaningful comparison, we must ensure that the classifier head does not significantly alter the structure between the embedding space and the output space. Since the spectral contrastive loss depends on the inner product between embeddings, a natural notion is to require that inner products be preserved, namely $\langle \boldsymbol{z}_1, \boldsymbol{z}_2 \rangle = \langle \boldsymbol{W}_H \boldsymbol{z}_1, \boldsymbol{W}_H \boldsymbol{z}_2 \rangle$ for all $\boldsymbol{z}_1, \boldsymbol{z}_2 \in \mathbb{R}^m$. Indeed, such transformations are called isometries and we will consider the case that $\boldsymbol{W}_H$ is an isometry in our analysis. [2] Note that for isometries to exist, the following assumption is necessary:

**Assumption 4.1.** The dimension of the embedding space is less than or equal to the number of original examples (i.e. the number of distinct labels). That is, $m \leq n$.

Fortunately this assumption is completely natural in the setting of DIET. Moreover, if Assumption 4.1 is satisfied, then requiring that $\boldsymbol{W}_H$ be an isometry does not restrict the expressivity of the model class since any model can be converted into an equivalent one where $\boldsymbol{W}_H$ is an isometry:

**Lemma 4.2.** *Suppose Assumption 4.1 holds and $f$ is a linear model $f_{\boldsymbol{W}}(\boldsymbol{x}) = \boldsymbol{W}\boldsymbol{x}$ and $\boldsymbol{W}_H$ is the projection head. For any model $(\boldsymbol{W}_H, \boldsymbol{W})$, there exists another model $(\boldsymbol{W}_H', \boldsymbol{W}')$ such that the model outputs agree, i.e. $\boldsymbol{W}_H \boldsymbol{W} = \boldsymbol{W}_H' \boldsymbol{W}'$, and $\boldsymbol{W}_H'$ is an isometry.*

In this setting we find that the minimizers of the spectral contrastive loss and MSE-DIET loss are equivalent:

**Theorem 4.3.** *Suppose that Assumption 4.1 holds and $f$ is a linear model $f_{\boldsymbol{W}}(\boldsymbol{x}) = \boldsymbol{W}\boldsymbol{x}$. Then,*

- *If $(\boldsymbol{W}, \boldsymbol{W}_H)$ is a global minimizer of $\mathcal{L}_{diet}^{mse}$ and $\boldsymbol{W}_H$ is an isometry, then $\boldsymbol{W}$ is a global minimizer of $\mathcal{L}_{scl}$.*

- *If $\boldsymbol{W}$ is a global minimizer of $\mathcal{L}_{scl}$, then there exists $\boldsymbol{W}_H$ such that $\boldsymbol{W}_H$ is an isometry and $(\boldsymbol{W}, \boldsymbol{W}_H)$ is a global minimizer of $\mathcal{L}_{diet}^{mse}$.*

The proofs are presented in Appendix B.1 and B.2. The previous theorem shows that the complicated contrastive loss is unnecessary; the same embedding structure can be induced using a simple supervised loss. The result is surprising given that the supervised loss is just the average of the loss computed independently for each example, while the contrastive loss explicitly computes pairwise similarities between examples.

Although the exact equivalence between global minima does not hold for nonlinear models, we empirically show in our experiments and further provide evidence in Appendix D that the high level structure of embeddings produced by DIET and CL are still remarkably similar in practice.

## 5 IMPROVING FEATURE LEARNING WITH DIET

While being theoretically on par with CL, we still find other limitations with DIET, including failing to learn relevant features and a large memory footprint. In this section, we address these shortcomings.

---

[2]In practice, Xue et al. (2024) showed that a linear projection head performs simple feature rescaling, a phenomenon related to neural collapse Papyan et al. (2020). So we expect our result to hold up to rescaling.

## 5.1 LEARNING MORE FEATURES

First, we investigate a failure mode of DIET and theoretically show that it cannot learn noisier features that might be relevant to downstream tasks. Ideally, we would like the SSL representations to capture as many semantically meaningful features of the input, as any of them could be useful for a given downstream task. For example, if we learn representations on images of dogs and the downstream task is to predict the dog breed, some species may be more easily differentiated by the color of their hair or fur, while others may be distinguished by the shape of their ears or the length of their tail. Then, we show that normalizing representations before applying the classifier head can address this limitation.

**Setting.** Let $\mathcal{C} = \{1, \ldots, C\}$ label a set of latent concepts. To each $c \in \mathcal{C}$ we assign a low noise feature $\boldsymbol{u}_c$ and a high noise feature $\boldsymbol{v}_c$. We assume all $\boldsymbol{u}_i$ and $\boldsymbol{v}_i$ are orthonormal. Let $\mathcal{G}_1, \mathcal{G}_2$ be noise distributions. Every training example takes the form

$$\boldsymbol{x} = (1 + \epsilon_1)\boldsymbol{u}_c + (1 + \epsilon_2)\boldsymbol{v}_c + \boldsymbol{\xi},$$

where $c \in \mathcal{C}$ and $\epsilon_1 \sim \mathcal{G}_1, \epsilon_2 \sim \mathcal{G}_2, \boldsymbol{\xi} \sim \mathcal{N}\left(\boldsymbol{0}, \frac{\phi^2}{d}\left(\boldsymbol{I}_d - \boldsymbol{u}_c\boldsymbol{u}_c^\top - \boldsymbol{v}_c\boldsymbol{v}_c^\top\right)\right)$.

We assume that $\mathcal{G}_1$ and $\mathcal{G}_2$ are symmetric with zero mean and variance $\sigma_1^2$ and $\sigma_2^2$, respectively with $\sigma_1^2 < \sigma_2^2$, and that $\mathcal{G}_1$ and $\mathcal{G}_2$ have absolute value bounded by some $\nu_1, \nu_2 < 1$, respectively. We define our data augmentation $A$ as that which replaces the noise components $\epsilon_1, \epsilon_2, \boldsymbol{\xi}$ with fresh noise drawn from the same distribution. This data model is a variant of the sparse coding model that is common in the feature learning literature (Wen & Li, 2021; Zou et al., 2021; Chen et al., 2023; Xue et al., 2023).

Given $n$ examples $\boldsymbol{x}_1, \ldots, \boldsymbol{x}_n \in \mathbb{R}^d$ distributed equally across the classes and a linear encoder $f_{\boldsymbol{W}}(\boldsymbol{x}) = \boldsymbol{W}\boldsymbol{x}$, we can consider minimizing the original DIET loss:

$$\mathcal{L}_{diet}^{mse} = \frac{1}{2n} \sum_{i=1}^{n} \mathbb{E}_A[\|\boldsymbol{W}_H \boldsymbol{W}(A(\boldsymbol{x}_i)) - \boldsymbol{e}_i\|^2], \tag{4}$$

or the normalized DIET loss, where we normalize representations before applying the classifier head:

$$\mathcal{L}_{diet-norm}^{mse} = \frac{1}{2n} \sum_{i=1}^{n} \mathbb{E}_A[\|\boldsymbol{W}_H(norm(\boldsymbol{W}(A(\boldsymbol{x}_i)))) - \boldsymbol{e}_i\|^2.$$

We also make the following technical assumptions:

1. Isometric classifier head: $\boldsymbol{W}_H$ is a fixed isometry. As before, this allows us to study the structure of the embedding space induced by the loss function without worrying about the effect of $\boldsymbol{W}_H$.
2. Alignment: For all $i, h_i = \|\boldsymbol{W}_H^\top \boldsymbol{e}_i\| \neq 0$. If $h_i = 0$, then the model outputs would always be perpendicular to $\boldsymbol{e}_i$, so the normalized DIET loss on $\boldsymbol{x}_i$ would be constant. Requiring $h_i \neq 0$ ensures that $\boldsymbol{x}_i$ can contribute to the learning.
3. Initialization: We initialize $\boldsymbol{W} = \boldsymbol{0}$, and train using gradient descent on the population loss.
4. Sparse concepts: $|C| = o(d)$.

### 5.1.1 NORMALIZED DIET LEARNS FEATURES MORE EQUALLY

In the above setting we prove that normalized DIET can capture both features when DIET cannot:

**Theorem 5.1.** *If $\boldsymbol{W}$ is a minimizer of $\mathcal{L}_{diet}^{mse}$ obtained from the above procedure, then*

$$\frac{\|\boldsymbol{W}\boldsymbol{v}_c\|}{\|\boldsymbol{W}\boldsymbol{u}_c\|} = \frac{\sigma_1^2}{\sigma_2^2} + o(1)$$

*On the other hand, if $\boldsymbol{W}$ is a minimizer of $\mathcal{L}_{diet-norm}^{mse}$ obtained from the above procedure, then*

$$\frac{1-\nu_1}{1+\nu_2} \leq \frac{\|\boldsymbol{W}\boldsymbol{v}_c\|}{\|\boldsymbol{W}\boldsymbol{u}_c\|} \leq \frac{1+\nu_1}{1-\nu_2}$$

The proof is detailed in Appendix B.3. The previous theorem shows that if $\sigma_1^2$ is much smaller than $\sigma_2^2$ then the alignment of the weight matrix $\boldsymbol{W}$ with the feature $\boldsymbol{v}_c$ will be small. In other words, DIET may fail to learn a feature if there is a less noisy feature present. On the other hand, normalized DIET will learn both features approximately equally so long as the noise does not significantly corrupt the feature. For example, if the noise ratio is bounded by $\nu_1, \nu_2 \leq \frac{1}{2}$, then the alignment of the weight matrix with the noisy feature and the clean feature will differ by at most a factor of 3. We perform extensive experiments in Section 6.1 to validate the effect of normalization in practical settings.

**Connections with Contrastive Learning.** Recall that the contrastive loss (Eq. 3) is computed based on the cosine similarity between representations, which depends only on the normalized representations. Chen et al. (2020) showed that applying normalization improves performance empirically, but as of yet theoretical characterizations of the effect of normalization are largely unstudied in the setting of SSL. Our result demonstrates provable benefits for DIET: using normalized representations during training can alleviate a failure mode whereby the learning of one feature is suppressed by the learning of another, less noisy feature. We expect the insights from our analysis are also applicable to other SSL methods.

### 5.1.2 PROJECTION HEAD ALSO IMPROVES FEATURE LEARNING

A separate line of work studies the benefits of a projection head in CL, which is added to the model during training but discarded during evaluation (Chen et al., 2020; Gupta et al., 2022). (Xue et al., 2024) show that the insertion of a projection head leads to more balanced feature learning in the embedding space. Notably, the analysis can also be applied to supervised learning, and hence to DIET. We confirm experimentally in Section 6.1 that projection head also improves the performance of DIET.

### 5.2 MAKING DIET MEMORY-EFFICIENT

Our last step is to reduce the memory requirements of DIET. Recall that the classifier head $\boldsymbol{W}_H$ grows linearly in the number of training examples $n$, making it impractical to load the entire classifier head into GPU memory for larger datasets.

### 5.2.1 BATCH CROSS ENTROPY

Our key observation is that the gradients due to logits corresponding to labels that do not appear in a batch do not contribute significantly to the gradient of the batch. Formally, given a batch of indices $\mathcal{I} \subset [n]$ with $|\mathcal{I}| = b$, let $\boldsymbol{X}_{\mathcal{I}}$ collect the corresponding input examples, and let $\boldsymbol{W}_H[\mathcal{I}] \in \mathbb{R}^{b \times m}$ collect the $i$-th row of $\boldsymbol{W}_H$ for $i \in \mathcal{I}$. Also let $A \in \mathcal{A}$ be some data augmentation. We hypothesize that

$$\nabla_\theta CE_n(\boldsymbol{W}_H f_\theta(A(\boldsymbol{X}_{\mathcal{I}})), \mathcal{I}) \approx \nabla_\theta CE_b(\boldsymbol{W}_H[\mathcal{I}] f_\theta(A(\boldsymbol{X}_{\mathcal{I}})), [0, \ldots, b-1]). \quad (5)$$

On the LHS is the standard cross entropy loss performed on the outputs of the classifier head, which requires cross entropy on $n$-dimensional vectors. On the RHS, we select only the $b$ rows corresponding to the indices found in the batch. We then reassign each example a distinct label from $\{0, \ldots, b-1\}$ and perform $b$-dimensional cross entropy. We call this *batch cross entropy*. We validate the above approximation empirically in Section 6.1.2.

The key point is that the RHS of Eq. 5 only requires $b$ rows of the classifier head to be loaded into memory at any point to calculate the forward and backward pass, while the full $n$ rows can be kept in high-capacity

storage. Note that in the standard case that the embedding dimension $m$ is much smaller than the input dimension $d$, loading the relevant rows of the classifier head and any corresponding information for the optimizer requires only a fraction of the cost of loading the input data.

### 5.2.2 HANDLING STATEFUL OPTIMIZERS

One more optimization can be made for per-parameter-stateful optimizers such as SGD with momentum or AdamW (Loshchilov & Hutter, 2019), as we will discuss next.

**Momentum.** Recall the update rule of SGD with learning rate $\eta$, momentum $\mu$, dampening $\tau$, weight decay $\lambda$:

$$\boldsymbol{m}_t \longleftarrow \mu \boldsymbol{m}_{t-1} + \tau \boldsymbol{g}_t, \qquad \boldsymbol{\theta}_t \longleftarrow (1 - \eta\lambda)\boldsymbol{\theta}_{t-1} - \eta \boldsymbol{m}_t \qquad (6)$$

Observe that the optimizer may update the weights even if their gradient at the current step $\boldsymbol{g}_t$ is zero. Thus using batch cross entropy would still require updating the entire classifier head at every step. To improve this, note that the $i$-th row of $\boldsymbol{W}_H$ is used only when the $i$-th example is selected in a batch; otherwise the gradient of batch cross entropy will be zero. Therefore when we encounter the $i$-th example, we can perform $t$ steps of optimizer updates on the $i$-th row of $\boldsymbol{W}_H$ immediately, where $t$ is the number of steps until the next time the $i$-th example will be chosen. Although we may not know $t$ exactly due to the randomness of minibatch sampling, we can make a simple estimate $t = \frac{N}{b}$, the size of the training dataset divided by the batch size. Note that apart from the first step, the remaining $t - 1$ steps all apply an update using zero gradient. These $t - 1$ steps can often be performed much more efficiently than directly running the optimizer for $t - 1$ steps. For example, if $\boldsymbol{g}_t = 0$ for all $t$, the above update formulas for SGD with momentum become an inhomogeneous linear recurrence relation which has a closed form solution:

$$\boldsymbol{m}_t = \mu^t \boldsymbol{m}_0, \qquad \boldsymbol{\theta}_t = (1 - \eta\lambda)^t \boldsymbol{\theta}_0 + \frac{(1 - \eta\lambda)^t \eta\mu - \eta\mu^{t+1}}{1 - \eta\lambda - \mu} \boldsymbol{m}_0 \qquad (7)$$

To summarize, at each step we only update the weights and optimizer state of rows of $\boldsymbol{W}_H$ that were selected for batch cross entropy at that step. We perform the update by first taking one step using Eq. 6 with $\boldsymbol{g}_t$ as the calculated gradient, and then apply the update given by Eq. 7 for $t = \frac{N}{b} - 1$. We call the complete procedure the *multistep update formula* for SGD with momentum.

**AdamW.** Similarly, we can adapt a more complex optimizer such as AdamW, with the update rules:

$$\boldsymbol{m}_t \leftarrow \beta_1 \boldsymbol{m}_{t-1} + (1 - \beta_1)\boldsymbol{g}_t, \qquad \boldsymbol{v}_t \leftarrow \beta_2 \boldsymbol{v}_{t-1} + (1 - \beta_2)\boldsymbol{g}_t^2,$$

$$\boldsymbol{\theta}_t \leftarrow (1 - \eta\lambda)\boldsymbol{\theta}_{t-1} - \eta \frac{\sqrt{1 - \beta_2^t}}{1 - \beta_1^t} \frac{\boldsymbol{m}_t}{\sqrt{\boldsymbol{v}_t} + \epsilon}.$$

We consider a slightly simplified version in which we remove the term $\frac{\sqrt{1-\beta_2^t}}{1-\beta_1^t}$ from the update rule. For default settings $\beta_1 = 0.9, \beta_2 = 0.999$, this can be interpreted as a type of learning rate warmup. Assuming $\epsilon$ is negligible, the new recurrence relation for $\boldsymbol{\theta}_t$ can be simplified to the same form as SGD with momentum by considering the ratio $\frac{\boldsymbol{m}_t}{\sqrt{\boldsymbol{v}_t}}$ and setting $\mu = \frac{\beta_1}{\sqrt{\beta_2}}$. This gives:

$$\boldsymbol{m}_t = \beta_1^t \boldsymbol{m}_0, \qquad \boldsymbol{v}_t = \beta_2^t \boldsymbol{v}_0, \qquad \boldsymbol{\theta}_t = (1 - \eta\lambda)^t \boldsymbol{\theta}_0 + \frac{(1 - \eta\lambda)^t \eta\mu - \eta\mu^{t+1}}{1 - \eta\lambda - \mu} \frac{\boldsymbol{m}_0}{\sqrt{\boldsymbol{v}_0} + \epsilon}. \qquad (8)$$

### 5.2.3 PUTTING IT ALL TOGETHER: S-DIET

Leveraging our findings from the previous sections, we develop S-DIET, by making the following modification to DIET: (1) following our theoretical results in Section 5.1.1 we normalize the outputs of the projection head before applying the classifier head. (2) inspired by contrastive learning methods, we include a projection head on top of the embeddings but before the classifier head (3) to reduce the memory footprint of DIET, we use batch cross entropy and the multistep update formula for AdamW (Section 5.2.2) to update the classifier head. Full pseudocode for S-DIET is presented in Appendix E.

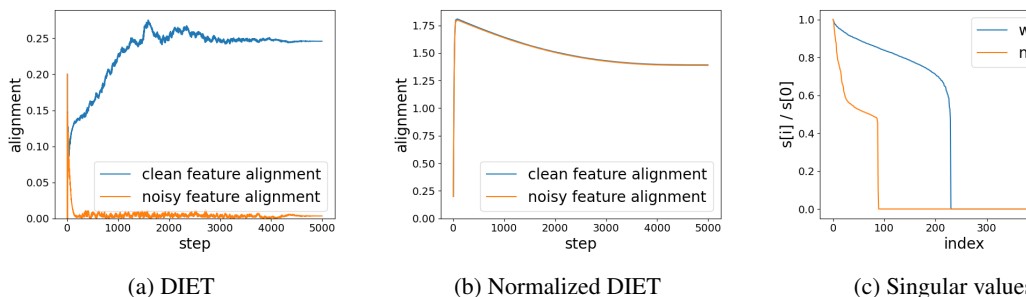

(a) DIET

(b) Normalized DIET

(c) Singular values

Figure 1: **(a) and (b) Alignment of weight matrix $W$** with clean feature $u_1$ and noisy feature $v_1$ (calculated as $\|Wu_1\|$ and $\|Wv_1\|$, respectively) when using DIET and normalized DIET, respectively. **(c) Singular values of representations** of the training dataset on CIFAR-100 taken before the classifier head (but after the projection head), sorted in decreasing order. Values are normalized by the largest singular value.

## 6 EXPERIMENTS

In this section, we validate the effectiveness and efficiency of S-DIET on a variety of datasets and models. First, we perform an in depth dive into the role of normalization in DIET through toy, synthetic, and real-world examples, bridging the gap between theory and practice. We also confirm the effectiveness of our Batch Cross Entropy, and the benefits of a projection head. Finally, we compare the performance of S-DIET with several contrastive baselines.

**Setting.** We perform experiments on a toy, a synthetic, and 4 real-world datasets: CIFAR-10, CIFAR-100 (Krizhevsky et al., 2009), ImageNet-100 (Tian et al., 2020), and TinyImageNet (Le & Yang, 2015). The CIFAR-10 and CIFAR-100 datasets consist of 50,000 training images and 10,000 test images drawn from 10 and 100 classes, respectively. ImageNet-100 contains of a subset of 100 classes from the ImageNet-1k dataset, consisting of almost 130,000 training examples. TinyImageNet contains 100,000 images from 200 classes at 64x64 resolution. For our models, we study the ResNet family of architectures, specifically ResNet-18 and ResNet-50 (He et al., 2016). We use a three layer ReLU MLP as a projection head during training. The rest of our experimental setup follows a unified setup from (Balestriero, 2023), as detailed in Appendix C.

### 6.1 BENEFITS OF NORMALIZATION, PROJECTION HEAD, AND BATCH CROSS ENTROPY

First, we confirm the effectiveness of each component of S-DIET. We start by verifying our theory on the benefits of normalization in Section 5.1.1 and its performance gain.

#### 6.1.1 S-DIET LEARNS MORE FEATURES THAN DIET

**Toy Example.** First, we instantiate the scenario from Section 5.1 with a more realistic training setup, showing that some of the more technical assumptions are not necessary. Specifically, we make the classifier head $W_H$ trainable from random initialization. In addition, instead of taking the expectation over all augmentations, we sample a single random augmentation of the input at each step. We also choose $\mathcal{G}_1, \mathcal{G}_2$ to follow normal distributions. Full experimental details are found in Appendix C.2.

Figure 1 show the resulting alignment between the clean and noisy feature of the first class. We observe that only the clean feature is learned with standard DIET, but both the clean and noisy features are learned almost equally when using normalization.

**Synthetic Dataset: MNIST on CIFAR-10.** Next, we construct a synthetic dataset where each input example consists of a CIFAR-10 image and an MNIST image of the same label concatenated along the

Table 1: **Linear Probe Accuracy** on synthetic dataset with and without masked MNIST digits.

| Normalize | No Masking | Masking |
|---|---|---|
| Yes | 83.9 | **84.06** |
| No | 13.76 | **43.56** |

Table 2: **Cosine similarity** of gradients for full cross entropy and batch cross entropy with randomly initialized models on CIFAR-100. We see that Batch CE closely approximates CE.

| Batch Size | Cosine Similarity | |
|---|---|---|
| | ResNet-18 | ResNet-50 |
| 64 | 0.9944 | 0.9960 |
| 128 | 0.9965 | 0.9980 |
| 256 | 0.9975 | 0.9990 |
| 512 | 0.9980 | 0.9995 |

Table 3: **Linear Probe Accuracy** of ResNet-18 on CIFAR-100 trained with and without normalization.

| Normalization | Accuracy |
|---|---|
| Yes | **66.88** |
| No | 62.60 |

Table 4: **Linear Probe Accuracy** on CIFAR-100 using embeddings from before and after the projection head.

| Model | Pre-projection | Post-projection |
|---|---|---|
| Resnet-18 | **66.88** | 63.46 |
| Resnet-50 | **72.34** | 67.60 |

channel dimension. We use weaker augmentations on the MNIST image, so that the MNIST image represents the clean feature and the CIFAR-10 image represents the noisy feature. Experimental details are found in Appendix C.3. We train a ResNet-18 using DIET with and without normalization. During linear probe evaluation, we may mask the MINST digit to compare how well the models learned the CIFAR image.

The results are shown in Table 1. We observe standard DIET quickly overfits the MNIST digit. Even when the MNIST digit is masked, linear probe performance is still poor, indicating that the CIFAR-10 features are not well learned. On the other hand, normalized DIET maintains high performance regardless of whether the MNIST digit is present, showing that the CIFAR-10 features are learned.

**Real-world Dataset: CIFAR-100.** On real world datasets such as CIFAR-100, it is difficult to determine what constitutes a clean or noisy feature. Instead, we propose to count the number of distinct features learned by the model. We use the number of large singular values of the representations of the training dataset as a proxy for the number of distinct learned features by the model. We define large singular values as those are at least some constant fraction $\alpha$ of the largest singular value, e.g. $\alpha = 0.1$. Due to the large dimension size of the output of the classifier head, we instead take representations from before the classifier head to compute the singular values. If normalized DIET indeed learns features more equally, then we can expect more large singular values from the representations of the model trained with normalization. Indeed, Figure 1c shows that DIET embeddings for CIFAR-100 have less than 100 large singular values, whereas there are over 200 large singular values when using normalization.

### 6.1.2 Ablation Study on S-DIET Components

**Normalization and Projection Head are Effective.** We perform additional experiments on CIFAR-100 to validate the effectiveness of normalization and the projection head in practice. First, we compare the linear probe accuracy when training with and without normalization. Indeed, in Table 3 we observe that training without normalization reduces the linear probe accuracy. Second, we check the performance of using the output of the projection head as the embeddings. Table 4 shows that using the outputs of the projection head performs worse than using the representations from before the projection head. These results are consistent with standard practice in contrastive learning.

**Batch Cross Entropy Closely Approximates Cross Entropy.** Next, we confirm that batch cross entropy closely approximates standard cross entropy by calculating the cosine similarity of gradients on CIFAR-100 for various batch sizes and randomly initialized models. We only consider parameters from the base model,

Table 5: **Linear Probe Accuracy** of S-DIET against DIET and various SSL baselines on CIFAR-10 and CIFAR-100 using a *batch size of 256*. [1]: results from (Balestriero, 2023).

| Method | CIFAR-10 | | CIFAR-100 | | ImageNet-100 | TinyImageNet |
| --- | --- | --- | --- | --- | --- | --- |
| | ResNet-18 | ResNet-50 | ResNet-18 | ResNet-50 | ResNet-50 | ResNet-50 |
| Barlow Twins | 90.38 | 90.62 | 66.84 | 68.06 | 79.52 | 45.20 |
| BYOL | 90.76 | 92.32 | 65.26 | 68.10 | (OOM) | 40.72 |
| SimCLR | 90.00 | 91.64 | 63.56 | 67.90 | 79.68 | 46.32 |
| Simsiam | 90.78 | 92.42 | 65.66 | 69.62 | 80.12 | 40.48 |
| DIET | 54.64 | 89.70 | 62.93[1] | 68.96[1] | 73.50[1] | 51.66[1] |
| S-DIET | **91.48** | **93.08** | **66.88** | **72.34** | **80.16** | **52.52** |

Table 6: **GPU Memory Usage** in MiB for S-DIET, DIET, and other SSL methods with a batch size of 256. OOM indicates out-of-memory on an Nvidia A40 GPU, which has 46068 MiB of memory.

| Method | CIFAR | | ImageNet-100 | Tiny-Imagenet |
| --- | --- | --- | --- | --- |
| | ResNet-18 | ResNet-50 | ResNet-50 | ResNet-50 |
| Barlow Twins | 4026 | 17090 | 44698 | 4532 |
| BYOL | 4512 | 17296 | (OOM) | 4842 |
| SimCLR | 3896 | 16408 | 40322 | 4352 |
| Simsiam | 3964 | 16562 | 45264 | 4390 |
| DIET | 2556 | 9720 | 31164 | 6676 |
| S-DIET | **2312** | **7770** | **23634** | **2976** |

not the projection head or classifier head. Table 2 shows that the cosine similarity between the gradients of batch cross entropy and full cross entropy is nearly 1 across different models and batch sizes, with higher cosine similarity for larger models and larger batch sizes.

## 6.2 S-DIET OUTPERFORMS DIET AND CONTRASTIVE BASELINES WITH LIMITED MEMORY

In Table 5, we compare the linear probe performance of S-DIET against DIET and SSL baselines when trained on CIFAR-10 and CIFAR-100 with batch size 256. We observe that S-DIET consistently outperforms DIET and SSL baselines with limited batch size. In addition, we highlight that DIET fails with the default hyperparameters on CIFAR-10 with ResNet-18 while S-DIET does not, indicating that S-DIET is less sensitive to changes in hyperparameters. Table 6 shows the corresponding memory usage of each methods. Due to the simplicity of the supervised loss compared to the complicated pairwise losses of other SSL methods, and optimizations around the classifier head, S-DIET has the minimum memory usage among the SSL methods.

## 7 CONCLUSION AND FUTURE WORK

Contrastive learning is among the most popular methods for self-supervised representation learning. Here, we rigorously analyzed the learning mechanism of an alternative approach, DIET. At first, it may seem that DIET's simple supervised architecture should not be able to match the performance of contrastive learning, which explicitly models pairwise interactions between examples. However, we derived a correspondence between the minima of DIET with MSE loss and the spectral contrastive loss. Then, we showed that normalizing the embeddings before applying the classifier head during training can prevent a failure mode where the learning of one less noisy feature suppresses the learning of another, noisier feature. We leveraged these observations, as well as improvements to DIET's memory consumption, to improve the performance of DIET to be on par with other SSL methods across a variety of datasets. Consequently, our modified DIET (S-DIET) presents a simple, effective, and memory-efficient solution for self-supervised representation learning. We believe that this inspires future work on this novel approach for representation learning.

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

## A    TECHNICAL CLARIFICATIONS

### A.1    NOTATION AND SETUP

We use regular font for scalars, bold lowercase font for vectors, bold uppercase font for matrices.

We use $\|\cdot\|$ to represent the Euclidean norm for vectors and $\|\cdot\|_F$ to represent the Frobenius norm for matrices. The vector $e_i$ represents the $i$-th standard basis vector. For a matrix $M$, we write $M^\dagger$ for the Moore-Penrose pseudoinverse of $M$.

We say a matrix $M \in \mathbb{R}^{m \times n}$ is an isometry if $M^\top M = I_m$. Equivalently, $\langle M v_1, M v_2 \rangle = \langle v_1, v_2 \rangle$ for all $v_1, v_2 \in \mathbb{R}^n$. We say $M$ is a partial isometry if $M$ acts as an isometry on the orthogonal complement of its kernel.

For a matrix $M \in \mathbb{R}^{m \times n}$ and a scalar function $g : \mathbb{R}^{m \times n} \to \mathbb{R}$, $\frac{\partial g}{\partial M}$ consists of the partial derivatives of $g$ with respect to the entries of $M$, namely

$$\frac{\partial g}{\partial M} = \begin{bmatrix} \frac{\partial g}{\partial M_{11}} & \cdots & \frac{\partial g}{\partial M_{1n}} \\ \vdots & \ddots & \vdots \\ \frac{\partial g}{\partial M_{m1}} & \cdots & \frac{\partial g}{\partial M_{mn}} \end{bmatrix}$$

We use the Kronecker delta function $\delta_{i,j}$, which is defined as 1 if $i = j$ otherwise 0.

### A.2    DEFINITION OF SPECTRAL CONTRASTIVE LOSS

Recall the given definition of the spectral contrastive loss

$$\mathcal{L}_{scl} = \mathbb{E}_{(\boldsymbol{x}_1, y_1), (\boldsymbol{x}_2, y_2) \sim \mathcal{D}} \left[ -\delta_{y_1, y_2} f(\boldsymbol{x}_1)^\top f(\boldsymbol{x}_2) \right] + \mathbb{E}_{(\boldsymbol{x}_1, y_1), (\boldsymbol{x}_2, y_2) \sim \mathcal{D}} \left[ (f(\boldsymbol{x}_1)^\top f(\boldsymbol{x}_2))^2 \right],$$

In Xue et al. (2023), the positive pair term in the contrastive loss was instead defined as

$$\mathbb{E}_{(x,y),(x,y') \sim \mathcal{D}, y=y'} \left[ -2 f(\boldsymbol{x})^\top f(\boldsymbol{x}') \right]$$

so that

$$\mathcal{L}_{scl}^* = \mathbb{E}_{(x,y),(x,y') \sim \mathcal{D}, y=y'} \left[ -2 f(\boldsymbol{x})^\top f(\boldsymbol{x}') \right] + \mathbb{E}_{(\boldsymbol{x}_1, y_1), (\boldsymbol{x}_2, y_2) \sim \mathcal{D}} \left[ (f(\boldsymbol{x}_1)^\top f(\boldsymbol{x}_2))^2 \right],$$

This only differs from the current definition by a some constant multiple $\alpha$, where $\alpha$ is the inverse of the probability that a randomly chosen pair is a positive pair. The reason for changing this normalization is that with the original formulation, the norm of the optimal weights and embeddings would grow with the number of classes. Quantitatively, it is not hard to check that

$$\mathcal{L}_{scl}^*(\alpha f) = \alpha^2 \mathcal{L}_{scl}(f)$$

That is, the loss landscape of the two loss functions is the same up to rescaling. It turns out this is the correct scaling factor to keep the norm of the optimal weights and embeddings bounded, with scale matching those produced by DIET.

### A.3 Normalization of Zero

Note that normalizing the zero vector is not well-defined. This can be an issue in the setup of Theorem 5.1 because we initialize $\boldsymbol{W} = \boldsymbol{0}$. In PyTorch, this is handled by redefining $norm(\boldsymbol{x}) \leftarrow \frac{\boldsymbol{x}}{\max\{\|\boldsymbol{x}\|, \epsilon\}}$ for negligible $\epsilon$. We will take a similar approach, where we simply define $norm(\boldsymbol{0}) = \boldsymbol{0}$ and the Jacobian as $\boldsymbol{J}_{norm}(\boldsymbol{0}) = \boldsymbol{I}$. This can be seen as taking $\epsilon \to 0$ and rescaling the Jacobian at $\boldsymbol{0}$ so that it does not blow up. Note that in the standard formula for the Jacobian of the normalization function,

$$\boldsymbol{J} = \frac{1}{\|\boldsymbol{x}\|}\left(\boldsymbol{I} - \frac{1}{\|\boldsymbol{x}^2\|}\boldsymbol{x}\boldsymbol{x}^\top\right)$$

the same formula holds when $\boldsymbol{x} = \boldsymbol{0}$ if we drop the $\|x\|$ terms. In the following proofs, this is how we will interpret such formulas in case we need to normalize a zero vector.

## B Proof of Theorems

### B.1 Proof of Lemma 4.2

*Proof.* Let $\boldsymbol{W}_H = \boldsymbol{U}\boldsymbol{\Sigma}\boldsymbol{V}^\top$ be an SVD of $\boldsymbol{W}_H$, where $\boldsymbol{U} \in \mathbb{R}^{n \times n}, \boldsymbol{\Sigma} \in \mathbb{R}^{n \times m}, \boldsymbol{V} \in \mathbb{R}^{m \times m}$. Since $\mathrm{k}\boldsymbol{W}_H \le m \le n$, this decomposition can be truncated so that

$$\boldsymbol{W}_H = \boldsymbol{U}_1\boldsymbol{\Sigma}_1\boldsymbol{V}^\top$$

where $\boldsymbol{U}_1 \in \mathbb{R}^{n \times m}, \boldsymbol{\Sigma}_1 \in \mathbb{R}^{m \times m}$ and $\boldsymbol{U}_1^\top\boldsymbol{U}_1 = \boldsymbol{I}_m$. Then taking $\boldsymbol{W}_H' = \boldsymbol{U}_1$ and $f' = \boldsymbol{\Sigma}_1\boldsymbol{V}^\top f$ works. $\square$

### B.2 Proof of Theorem 4.3

Denote by $N = |\mathcal{D}|$ be the size of the augmented dataset. We represent this dataset in matrix form

$$\mathcal{D} = (\boldsymbol{X}, \boldsymbol{Y}) \in \mathbb{R}^{d \times N} \times \mathbb{R}^{n \times N}$$

where every column of $\boldsymbol{X}$ is an augmented input and the corresponding column of $\boldsymbol{Y}$ is a one-hot encoding of the label.

Define the following useful matrices to characterize the structure of the data:

$$\boldsymbol{M} = \mathbb{E}_{(x,y) \sim \mathcal{D}}[\boldsymbol{x}\boldsymbol{x}^\top] = \frac{1}{N}\boldsymbol{X}\boldsymbol{X}^\top$$

$$\boldsymbol{M}_{pos} = \mathbb{E}_{(x_1,y_1),(x_2,y_2) \sim \mathcal{D}}[\boldsymbol{x}_1\boldsymbol{x}_2^\top \delta_{y_1,y_2}] = \frac{1}{N^2}\boldsymbol{X}\boldsymbol{Y}^\top\boldsymbol{Y}\boldsymbol{X}^\top$$

Here $\boldsymbol{M}$ is the expected outer product of all examples with themselves, and $\boldsymbol{M}_{pos}$ is the expected outer product between pairs of examples if they are in the same class (known as positive pairs).

We outline the proof as follows. First we leverage a result from Xue et al. (2023) which characterizes the critical points and global minima of the spectral contrastive loss in the same setting. We then prove a relationship between the critical points of MSE diet and the sepctral contrastive loss. Finally, we prove a relationship between the global minima of the two loss functions.

For the rest of this section, we will just write $\mathcal{L}_{diet}$ in place of $\mathcal{L}_{diet}^{mse}$.

The following is a statement and slightly simplified proof of the key theorem from Xue et al. (2023):

**Theorem B.1.** *A linear function $f(\boldsymbol{x}) = \boldsymbol{W}\boldsymbol{x}$ is a critical point of $\mathcal{L}_{scl}$ iff there is a basis such that*

$$\boldsymbol{M}^\dagger \boldsymbol{M}_{pos} = diag(\lambda_1, \dots, \lambda_r, \lambda_{r+1}, \dots, \lambda_d)$$
$$\boldsymbol{W}^\top \boldsymbol{W} \boldsymbol{M} = diag(\lambda_1, \dots, \lambda_r, 0, \dots, 0)$$
$$\boldsymbol{W}^\top \boldsymbol{W} \boldsymbol{M}_{pos} = diag(\lambda_1^2, \dots, \lambda_r^2, 0, \dots, 0)$$

*with $\lambda_1, \dots, \lambda_d \geq 0$ and we have $r \leq \operatorname{rank} \boldsymbol{W} \leq m$.*

*It is a global minimum of $\mathcal{L}_{scl}$ iff it satisfies*

$$\boldsymbol{W}^\top \boldsymbol{W} \boldsymbol{M} = [\boldsymbol{M}^\dagger \boldsymbol{M}_{pos}]_m$$

*Proof.* The first order condition for $\mathcal{L}_{scl}$

$$\frac{\partial \mathcal{L}_{scl}}{\partial \boldsymbol{W}} = -\boldsymbol{W} \boldsymbol{M}_{pos} + \boldsymbol{W} \boldsymbol{M} \boldsymbol{W}^\top \boldsymbol{W} \boldsymbol{M} = 0 \tag{9}$$

Since $\boldsymbol{M}$ and $\boldsymbol{M}_{pos}$ are positive semidefinite, $\boldsymbol{M}^\dagger \boldsymbol{M}_{pos}$ is diagonalizable. Therefore we can construct a basis $\{\boldsymbol{v_1}, \dots, \boldsymbol{v_d}\}$ of eigenvectors of $\boldsymbol{M}^\dagger \boldsymbol{M}_{pos}$ with corresponding eigenvalues $\lambda_1, \dots, \lambda_d$.

Now we have $\operatorname{im} \boldsymbol{M}_{pos} \subset \operatorname{im} \boldsymbol{M}$, which implies that $\boldsymbol{M}_{pos} = \boldsymbol{M} \boldsymbol{M}^\dagger \boldsymbol{M}_{pos}$. Then 9 implies that

$$(\boldsymbol{W}^\top \boldsymbol{W} \boldsymbol{M})^2 \boldsymbol{v}_i = \boldsymbol{W}^\top \boldsymbol{W} \boldsymbol{M} (\boldsymbol{M}^\dagger \boldsymbol{M}_{pos}) \boldsymbol{v}_i = \lambda_i \boldsymbol{W}^\top \boldsymbol{W} \boldsymbol{M} \boldsymbol{v}_i$$

Thus either $\boldsymbol{W}^\top \boldsymbol{W} \boldsymbol{M} \boldsymbol{v}_i = \boldsymbol{0}$ or $\boldsymbol{W}^\top \boldsymbol{W} \boldsymbol{M} \boldsymbol{v}_i$ is an eigenvector of $\boldsymbol{W}^\top \boldsymbol{W} \boldsymbol{M}$ with eigenvalue $\lambda_i$. Since $\boldsymbol{W}^\top \boldsymbol{W} \boldsymbol{M}$ is diagonalizable, the latter implies that $\boldsymbol{v}_i$ is also an eigenvalue of $\boldsymbol{W}^\top \boldsymbol{W} \boldsymbol{M}$ with $\boldsymbol{W}^\top \boldsymbol{W} \boldsymbol{M} \boldsymbol{v_i} = \lambda_i \boldsymbol{v_i} = \boldsymbol{M}^\dagger \boldsymbol{M}_{pos} \boldsymbol{v_i}$

Thus, with possible reordering of the $\boldsymbol{v}_i$, we have a basis $\boldsymbol{v}_1, \dots, \boldsymbol{v}_r, \dots, \boldsymbol{v}_d$ such that in this basis

$$\boldsymbol{M}^\dagger \boldsymbol{M}_{pos} = diag(\lambda_1, \dots, \lambda_r, \lambda_{r+1}, \dots, \lambda_d)$$
$$\boldsymbol{W}^\top \boldsymbol{W} \boldsymbol{M} = diag(\lambda_1, \dots, \lambda_r, 0, \dots, 0)$$
$$\boldsymbol{W}^\top \boldsymbol{W} \boldsymbol{M}_{pos} = diag(\lambda_1^2, \dots, \lambda_r^2, 0, \dots, 0)$$

with $\lambda_1, \dots, \lambda_d \geq 0$ and we have and $r \leq \operatorname{rank} \boldsymbol{W} \leq m$.

Note that if $\boldsymbol{W}$ admits the above form, then

$$\boldsymbol{W}^\top \boldsymbol{W} \boldsymbol{M}_{pos} = \boldsymbol{W}^\top \boldsymbol{W} \boldsymbol{M} \boldsymbol{W}^\top \boldsymbol{W} \boldsymbol{M}$$

which implies

$$\boldsymbol{W} \boldsymbol{M}_{pos} = \boldsymbol{W} \boldsymbol{M} \boldsymbol{W}^\top \boldsymbol{W} \boldsymbol{M}$$

hence all such $\boldsymbol{W}$ are critical points.

Then for all such $\boldsymbol{W}$,

$$
\begin{aligned}
\mathcal{L} &= \mathrm{Tr}[-2\boldsymbol{W}^\top\boldsymbol{W}\boldsymbol{M}_{pos} + \boldsymbol{W}^\top\boldsymbol{W}\boldsymbol{M}\boldsymbol{W}^\top\boldsymbol{W}\boldsymbol{M}] \\
&= -2\sum_{i=1}^{r}\lambda_i^2 + \sum_{i=1}^{r}\lambda_i^2 \\
&= -\sum_{i=1}^{r}\lambda_i^2
\end{aligned}
$$

It is clear from the above expression that the minimum among critical points is achieved when $r$ is maximal and $\lambda_1, \dots, \lambda_m$ are the largest eigenvalues. This happens if and only if

$$
\boldsymbol{W}^\top\boldsymbol{W}\boldsymbol{M} = [\boldsymbol{M}^\dagger\boldsymbol{M}_{pos}]_m
$$

It remains to check the behavior as $\|\boldsymbol{W}\|_F$ grows large. Equivalently, $\boldsymbol{W}^\top\boldsymbol{W}$ has a large eigenvalue $\lambda$. Let $\boldsymbol{w}$ be a corresponding eigenvector. If $\boldsymbol{w} \in \ker\boldsymbol{M}$, then $\boldsymbol{M}\boldsymbol{w} = \boldsymbol{M}_{pos}\boldsymbol{w} = 0$, so we see that the loss is unchanged. Otherwise, $\boldsymbol{w}$ has some nonzero alignment with $\mathrm{im}(\boldsymbol{W})$. But then $\mathrm{Tr}[\boldsymbol{W}^\top\boldsymbol{W}\boldsymbol{M}\boldsymbol{W}^\top\boldsymbol{W}\boldsymbol{M}]$ grows quadratically in $\lambda$, but $\mathrm{Tr}[-2\boldsymbol{W}^\top\boldsymbol{W}\boldsymbol{M}_{pos}]$ grows at most linearly in $\lambda$, hence the loss is large. We conclude that the previously found condition in fact specifies the global minimizers of $\mathcal{L}$. $\square$

The following lemma establishes a connection between the critical points of $\mathcal{L}_{diet}$ versus $\mathcal{L}_{scl}$.

**Lemma B.2.** *The following are true:*

- *If $(\boldsymbol{W}, \boldsymbol{W}_H)$ is a critical point of $\mathcal{L}_{diet}$ and $\boldsymbol{W}_H$ is an isometry, then $\boldsymbol{W}$ is a critical point of $\mathcal{L}_{scl}$.*

- *If $\boldsymbol{W}$ is a critical point of $\mathcal{L}_{scl}$, then there exists a partial isometry $\boldsymbol{W}_H$ such that $(\boldsymbol{W}, \boldsymbol{W}_H)$ is a critical point of $\mathcal{L}_{diet}$.*

*Proof.* The first order condition for $\mathcal{L}_{diet}$ requires that

$$
\frac{\partial\mathcal{L}_{diet}}{\partial\boldsymbol{W}} = \boldsymbol{W}_H^\top(\boldsymbol{W}_H\boldsymbol{W}\boldsymbol{X} - \boldsymbol{Y})\boldsymbol{X}^\top = 0 \tag{10}
$$

$$
\frac{\partial\mathcal{L}_{diet}}{\partial\boldsymbol{W}_H} = (\boldsymbol{W}_H\boldsymbol{W}\boldsymbol{X} - \boldsymbol{Y})\boldsymbol{X}^\top\boldsymbol{W}^\top = 0 \tag{11}
$$

On the other hand, the first order condition for $\mathcal{L}_{scl}$ is

$$
\boldsymbol{W}\boldsymbol{M}_{pos} = \boldsymbol{W}\boldsymbol{M}\boldsymbol{W}^\top\boldsymbol{W}\boldsymbol{M}.
$$

Indeed, if $\boldsymbol{W}$ is a critical point of $\mathcal{L}_{diet}$, then Equation 10 implies

$$
\boldsymbol{W}\boldsymbol{X}\boldsymbol{X}^\top = \boldsymbol{W}_H^\top\boldsymbol{Y}\boldsymbol{X}^\top \tag{12}
$$

And Equation 11 gives

$$
\boldsymbol{W}_H\boldsymbol{W}\boldsymbol{X}\boldsymbol{X}^\top\boldsymbol{W}^\top = \boldsymbol{Y}\boldsymbol{X}^\top\boldsymbol{W}^\top
$$

Taking transposes, we have

$$
\boldsymbol{W}\boldsymbol{X}\boldsymbol{X}^\top\boldsymbol{W}^\top\boldsymbol{W}_H^\top = \boldsymbol{W}\boldsymbol{X}\boldsymbol{Y}^\top \tag{13}
$$

Right multiplying by $\boldsymbol{W}_H$ and using the fact that $\boldsymbol{W}_H^\top \boldsymbol{W}_H = \boldsymbol{I}_m$ gives

$$\boldsymbol{W}\boldsymbol{X}\boldsymbol{X}^\top \boldsymbol{W}^\top = \boldsymbol{W}\boldsymbol{X}\boldsymbol{Y}^\top \boldsymbol{W}_H \tag{14}$$

Combining Equations 12 and 14, we get

$$\boldsymbol{W}\boldsymbol{X}\boldsymbol{X}^\top \boldsymbol{W}^\top \boldsymbol{W}\boldsymbol{X}\boldsymbol{X}^\top = \boldsymbol{W}\boldsymbol{X}\boldsymbol{Y}^\top \boldsymbol{W}_H \boldsymbol{W}_H^\top \boldsymbol{Y}\boldsymbol{X}^\top$$

We claim that

$$\boldsymbol{W}\boldsymbol{X}\boldsymbol{Y}^\top \boldsymbol{W}_H \boldsymbol{W}_H^\top = \boldsymbol{W}\boldsymbol{X}\boldsymbol{Y}^\top$$

Indeed, since $\boldsymbol{W}_H$ is an isometry, $\boldsymbol{W}_H^\top$ is a partial isometry, so $\boldsymbol{W}_H \boldsymbol{W}_H^\top$ has a basis $\{\boldsymbol{v}_1, \ldots, \boldsymbol{v}_n\}$ such that $\boldsymbol{W}_H \boldsymbol{W}_H^\top \boldsymbol{v}_i = \boldsymbol{v}_i$ or $\boldsymbol{W}_H \boldsymbol{W}_H^\top \boldsymbol{v}_i = \boldsymbol{0}$. If the former is true, then clearly $\boldsymbol{W}\boldsymbol{X}\boldsymbol{Y}^\top \boldsymbol{W}_H \boldsymbol{W}_H^\top \boldsymbol{v}_i = \boldsymbol{W}\boldsymbol{X}\boldsymbol{Y}^\top \boldsymbol{v}_i$. If the latter is true, then we know that $\boldsymbol{W}_H^\top \boldsymbol{v}_i = \boldsymbol{0}$. But then by Equation 13 we have

$$\boldsymbol{W}\boldsymbol{X}\boldsymbol{Y}^\top \boldsymbol{v}_i = \boldsymbol{W}\boldsymbol{X}\boldsymbol{X}^\top \boldsymbol{W}^\top \boldsymbol{W}_H^\top \boldsymbol{v}_i = \boldsymbol{0}$$

Since equality holds on a basis, we conclude the two matrix products are equal, as claimed.

Thus we now have

$$\boldsymbol{W}\boldsymbol{X}\boldsymbol{X}^\top \boldsymbol{W}^\top \boldsymbol{W}\boldsymbol{X}\boldsymbol{X}^\top = \boldsymbol{W}\boldsymbol{X}\boldsymbol{Y}^\top \boldsymbol{Y}\boldsymbol{X}^\top$$

Substituting the values $\boldsymbol{M} = \boldsymbol{X}\boldsymbol{X}^\top$ and $\boldsymbol{M}_{pos} = \boldsymbol{X}\boldsymbol{Y}^\top \boldsymbol{Y}\boldsymbol{X}$,

$$\boldsymbol{W}\boldsymbol{M}_{pos} = \boldsymbol{W}\boldsymbol{M}\boldsymbol{W}^\top \boldsymbol{W}\boldsymbol{M}$$

as desired.

For the converse, suppose that $\boldsymbol{W}$ is a critical point of $\mathcal{L}_{scl}$, namely

$$\boldsymbol{W}\boldsymbol{M}_{pos} = \boldsymbol{W}\boldsymbol{M}\boldsymbol{W}^\top \boldsymbol{W}\boldsymbol{M}$$

Let $V = \ker(\boldsymbol{M}_{pos} - \boldsymbol{M}\boldsymbol{W}^\top \boldsymbol{W}\boldsymbol{M})$. Since $\boldsymbol{M}_{pos} - \boldsymbol{M}\boldsymbol{W}^\top \boldsymbol{W}\boldsymbol{M}$ is symmetric, $V^\perp$ is spanned by eigenvectors with nonzero eigenvalues. Let $\boldsymbol{v}$ be such an eigenvector with eigenvalue $\lambda \neq 0$. Then

$$\boldsymbol{0} = \boldsymbol{W}(\boldsymbol{M}_{pos} - \boldsymbol{M}\boldsymbol{W}^\top \boldsymbol{W}\boldsymbol{M})\boldsymbol{v} = \lambda \boldsymbol{W}\boldsymbol{v}$$

It follows that $\boldsymbol{W}\boldsymbol{v} = 0$, so $V^\perp \subset \ker \boldsymbol{W}$.

Set $U = (\boldsymbol{W}\boldsymbol{X}\boldsymbol{X}^\top)(V)$, $Z = (\boldsymbol{Y}\boldsymbol{X}^\top)(V)$. Since

$$(\boldsymbol{Y}\boldsymbol{X}^\top)^\top(\boldsymbol{Y}\boldsymbol{X}^\top) = \boldsymbol{M}_{pos} = \boldsymbol{M}\boldsymbol{W}^\top \boldsymbol{W}\boldsymbol{M} = (\boldsymbol{W}\boldsymbol{X}\boldsymbol{X}^\top)^\top(\boldsymbol{W}\boldsymbol{X}\boldsymbol{X}^\top)$$

when restricted to $V$, there exists an isometry $\boldsymbol{W}_H' : U \to Z$ such that $\boldsymbol{Y}\boldsymbol{X}^\top = \boldsymbol{W}_H \boldsymbol{W}\boldsymbol{X}\boldsymbol{X}^\top$ on $V$ and $\boldsymbol{X}\boldsymbol{Y}^\top = \boldsymbol{X}\boldsymbol{X}^\top \boldsymbol{W}^\top \boldsymbol{W}_H^\top$ on $Z$. Extend $\boldsymbol{W}_H'$ to a partial isometry $\boldsymbol{W}_H : \mathbb{R}^m \to \mathbb{R}^n$ such that $\boldsymbol{W}_H|_U = \boldsymbol{W}_H'$ and $\boldsymbol{W}_H|_{U^\perp} = \boldsymbol{0}$.

Now using the fact that $\operatorname{im}(\boldsymbol{W}^\top) = \ker(\boldsymbol{W})^\perp \subset V$, we have

$$\boldsymbol{Y}\boldsymbol{X}^\top \boldsymbol{W}^\top = \boldsymbol{W}_H \boldsymbol{W}\boldsymbol{X}\boldsymbol{X}^\top \boldsymbol{W}^\top$$

Also

$$\boldsymbol{X}\boldsymbol{Y}^\top \boldsymbol{W}_H = \boldsymbol{X}\boldsymbol{X}^\top \boldsymbol{W}^\top \boldsymbol{W}_H^\top \boldsymbol{W}_H$$

because any vector in $\mathbb{R}^m$ can be written as $\boldsymbol{u} + \boldsymbol{u}_\perp$ where $\boldsymbol{u} \in U, \boldsymbol{u}_\perp \in U^\perp$ and

$$\begin{aligned}
\boldsymbol{X}\boldsymbol{Y}^\top \boldsymbol{W}_H(\boldsymbol{u} + \boldsymbol{u}_\perp) &= \boldsymbol{X}\boldsymbol{Y}^\top \boldsymbol{W}_H \boldsymbol{u} \\
&= \boldsymbol{X}\boldsymbol{X}^\top \boldsymbol{W}^\top \boldsymbol{W}_H^\top \boldsymbol{W}_H \boldsymbol{u} \\
&= \boldsymbol{X}\boldsymbol{X}^\top \boldsymbol{W}^\top \boldsymbol{W}_H^\top \boldsymbol{W}_H(\boldsymbol{u} + \boldsymbol{u}_\perp)
\end{aligned}$$

These are the two conditions for being a critical point of $\mathcal{L}_{diet}$, completing the proof.

$\square$

We now narrow our attention from critical points to global minima. The above Lemma means that we can restrict our study to the critical points of $\mathcal{L}_{scl}$. Using this fact, we can now characterize the global minimizers of $\mathcal{L}_{diet}$ as follows:

**Theorem B.3.** *Assume that $\boldsymbol{W}_H$ is an isometry. Then $(\boldsymbol{W}, \boldsymbol{W}_H)$ is global minimizer of $\mathcal{L}_{diet}$ iff the following hold*

$$\boldsymbol{W}^\top \boldsymbol{W} \boldsymbol{M} = [\boldsymbol{M}^\dagger \boldsymbol{M}_{pos}]_m$$

$$\frac{1}{N} \operatorname{Tr}(\boldsymbol{W} \boldsymbol{X} \boldsymbol{Y}^\top \boldsymbol{W}_H^\top) = \operatorname{Tr}([\boldsymbol{M}^\dagger \boldsymbol{M}_{pos}]_m)$$

*Proof.* Suppose $(\boldsymbol{W}, \boldsymbol{W}_H)$ is a global minimizer of $\mathcal{L}_{diet}$ and $\boldsymbol{W}_H$ is an isometry. By Lemma B.2, $\boldsymbol{W}$ is a critical point of $\mathcal{L}_{scl}$. By Theorem B.1, there is a basis such that

$$\boldsymbol{M}^\dagger \boldsymbol{M}_{pos} = diag(\lambda_1, \ldots, \lambda_r, \lambda_{r+1}, \ldots, \lambda_d)$$
$$\boldsymbol{W}^\top \boldsymbol{W} \boldsymbol{M} = diag(\lambda_1, \ldots, \lambda_r, 0, \ldots, 0)$$
$$\boldsymbol{W}^\top \boldsymbol{W} \boldsymbol{M}_{pos} = diag(\lambda_1^2, \ldots, \lambda_r^2, 0, \ldots, 0)$$

with $\lambda_1, \ldots, \lambda_d \geq 0$ and we have $r \leq \operatorname{rank} \boldsymbol{W} \leq m$.

Now calculating the value of the loss

$$\begin{aligned}
\mathcal{L}_{diet} &= \frac{1}{2} \mathbb{E}_{\mathcal{D}}[\|\boldsymbol{W}_H \boldsymbol{W} \boldsymbol{x}_i - e_{y_i}\|^2] \\
&= \frac{1}{2N} \|\boldsymbol{W}_H \boldsymbol{W} \boldsymbol{X} - \boldsymbol{Y}\|_F^2 \\
&= \frac{1}{2N} \operatorname{Tr}((\boldsymbol{W}_H \boldsymbol{W} \boldsymbol{X} - \boldsymbol{Y})^\top (\boldsymbol{W}_H \boldsymbol{W} \boldsymbol{X} - \boldsymbol{Y})) \\
&= \frac{1}{2N} \operatorname{Tr}(\boldsymbol{X}^\top \boldsymbol{W}^\top \boldsymbol{W}_H^\top \boldsymbol{W}_H \boldsymbol{W} \boldsymbol{X} - \boldsymbol{X}^\top \boldsymbol{W}^\top \boldsymbol{W}_H^\top \boldsymbol{Y} - \boldsymbol{Y}^\top \boldsymbol{W}_H \boldsymbol{W} \boldsymbol{X} + \boldsymbol{Y}^\top \boldsymbol{Y}) \\
&= \frac{1}{2N} \left( \operatorname{Tr}(\boldsymbol{W}^\top \boldsymbol{W} \boldsymbol{X} \boldsymbol{X}^\top) - 2 \operatorname{Tr}(\boldsymbol{W} \boldsymbol{X} \boldsymbol{Y}^\top \boldsymbol{W}_H) + \operatorname{Tr}(\boldsymbol{Y}^\top \boldsymbol{Y}) \right)
\end{aligned}$$

Observe that

$$\frac{1}{N} \operatorname{Tr}(\boldsymbol{W}^\top \boldsymbol{W} \boldsymbol{X} \boldsymbol{X}^\top) = \operatorname{Tr}(\boldsymbol{W}^\top \boldsymbol{W} \boldsymbol{M}) = \sum_{i=1}^r \lambda_i$$

Also $\boldsymbol{W}^\top \boldsymbol{W} \boldsymbol{M}_{pos} = \frac{1}{N^2} \boldsymbol{W}^\top \boldsymbol{W} \boldsymbol{X} \boldsymbol{Y}^\top \boldsymbol{Y} \boldsymbol{X}^\top$ and $\frac{1}{N^2} \boldsymbol{W} \boldsymbol{X} \boldsymbol{Y}^\top \boldsymbol{Y} \boldsymbol{X}^\top \boldsymbol{W}^\top$ are diagonalizable and have the same nonzero eigenvalues, namely $\lambda_1^2, \ldots, \lambda_r^2$. Using the fact that

$$\boldsymbol{W} \boldsymbol{X} \boldsymbol{Y}^\top \boldsymbol{W}_H \boldsymbol{W}_H^\top = \boldsymbol{W} \boldsymbol{X} \boldsymbol{Y}^\top$$

we have

$$(\frac{1}{N} \boldsymbol{W} \boldsymbol{X} \boldsymbol{Y}^\top \boldsymbol{W}_H)(\frac{1}{N} \boldsymbol{W} \boldsymbol{X} \boldsymbol{Y}^\top \boldsymbol{W}_H)^\top = \frac{1}{N^2} \boldsymbol{W} \boldsymbol{X} \boldsymbol{Y}^\top \boldsymbol{Y} \boldsymbol{X}^\top \boldsymbol{W}^\top,$$

we conclude by the Spectral Theorem that

$$\frac{1}{N} \operatorname{Tr}(\boldsymbol{W} \boldsymbol{X} \boldsymbol{Y}^\top \boldsymbol{W}_H) \leq \sum_{i=1}^r \lambda_i \tag{15}$$

Finally, note that $\operatorname{Tr}(\boldsymbol{Y}^\top \boldsymbol{Y})$ is a constant. Therefore the minimum possible value of the loss is when $\boldsymbol{W}^\top \boldsymbol{W} \boldsymbol{M} = [\boldsymbol{M}^\dagger \boldsymbol{M}_{pos}]_m$ and equality holds in equation 15 with $r = m$ and $\lambda_1, \ldots, \lambda_m$ the $m$ largest eigenvalues of $\boldsymbol{M}^\dagger \boldsymbol{M}_{pos}$. It only remains to show this value of the loss is achievable.

Indeed, it is not hard to find $\boldsymbol{W}$ such that $\boldsymbol{W}^\top \boldsymbol{W} \boldsymbol{M} = [\boldsymbol{M}^\dagger \boldsymbol{M}_{pos}]_m$ (for example take a global minimizer of $\mathcal{L}_{scl}$).

Let $\boldsymbol{W} \boldsymbol{X} \boldsymbol{Y}^\top = \boldsymbol{U} \boldsymbol{\Sigma} \boldsymbol{V}^\top$ be a singular value decomposition of $\boldsymbol{W} \boldsymbol{X} \boldsymbol{Y}^\top$. Let $\boldsymbol{W}_H : \mathbb{R}^m \to \mathbb{R}^n$ map the ith eigenvector of $\boldsymbol{U}$ to the ith eigenvector of $\boldsymbol{V}$ for $i = 1, \ldots, p$. Then

$$\boldsymbol{W} \boldsymbol{X} \boldsymbol{Y}^\top \boldsymbol{W}_H = \boldsymbol{U} \boldsymbol{\Sigma} \boldsymbol{U}^\top.$$

In particular, $\boldsymbol{W} \boldsymbol{X} \boldsymbol{Y}^\top \boldsymbol{W}_H$ is a positive semidefinite matrix, and

$$\frac{1}{N^2} (\boldsymbol{W} \boldsymbol{X} \boldsymbol{Y}^\top \boldsymbol{W}_H)^2 = \frac{1}{N^2} \boldsymbol{W} \boldsymbol{X} \boldsymbol{Y}^\top \boldsymbol{Y} \boldsymbol{X}^\top \boldsymbol{W}^\top$$

has nonzero eigenvalues $\lambda_1^2, \ldots, \lambda_r^2$, so $\frac{1}{N} \boldsymbol{W} \boldsymbol{X} \boldsymbol{Y}^\top \boldsymbol{W}_H$ has eigenvalues $\lambda_1, \ldots, \lambda_r$. Thus $\frac{1}{N} \operatorname{Tr}(\boldsymbol{W} \boldsymbol{X} \boldsymbol{Y}^\top \boldsymbol{W}_H) = \sum_{i=1}^r \lambda_i$ and $(\boldsymbol{W}, \boldsymbol{W}_H)$ as constructed achieves the minimum value of $\mathcal{L}_{diet}$. This completes the proof. $\square$

With the above two results, we obtain the desired result:

**Theorem 4.3.** *Suppose that Assumption 4.1 holds and $f$ is a linear model $f_{\boldsymbol{W}}(\boldsymbol{x}) = \boldsymbol{W} \boldsymbol{x}$. Then,*

- *If $(\boldsymbol{W}, \boldsymbol{W}_H)$ is a global minimizer of $\mathcal{L}_{diet}^{mse}$ and $\boldsymbol{W}_H$ is an isometry, then $\boldsymbol{W}$ is a global minimizer of $\mathcal{L}_{scl}$.*

- *If $\boldsymbol{W}$ is a global minimizer of $\mathcal{L}_{scl}$, then there exists $\boldsymbol{W}_H$ such that $\boldsymbol{W}_H$ is an isometry and $(\boldsymbol{W}, \boldsymbol{W}_H)$ is a global minimizer of $\mathcal{L}_{diet}^{mse}$.*

*Proof.* The first claim is immediate from Theorems B.1 and B.3. For the second claim, we in fact constructed the necessary $\boldsymbol{W}_H$ in the proof of Theorem B.3. $\square$

### B.3 Proof of Theorem 5.1

We will first prove the claim about $\mathcal{L}_{diet}$. Then we will prove the claim about $\mathcal{L}_{diet-norm}^{mse}$ in a sequence of lemmas.

**Lemma B.4.** *If $\boldsymbol{W}$ is a minimizer of $\mathcal{L}_{diet}^{mse}$ as defined in Equation 4, then*

$$\frac{\|\boldsymbol{W} \boldsymbol{v}_c\|}{\|\boldsymbol{W} \boldsymbol{u}_c\|} = \frac{\sigma_1^2}{\sigma_2^2} + o(1)$$

*Proof.* Since $\boldsymbol{W}_H$ is fixed, minimizing $\mathcal{L}_{diet}$ is in fact just standard linear regression. The closed form solution is well known:

$$\boldsymbol{W} = \boldsymbol{W}_H^\top \left( \frac{1}{n} \sum_{i=1}^n \mathbb{E}_A[\boldsymbol{e}_i A(\boldsymbol{x}_i)^\top] \right) \left( \frac{1}{n} \sum_{i=1}^n \mathbb{E}_A[A(\boldsymbol{x}_i) A(\boldsymbol{x}_i)^\top] \right)^{-1} \tag{16}$$

Now we calculate

$$\mathbb{E}_A[A(\boldsymbol{x}_i) A(\boldsymbol{x}_i)^\top] = \mathbb{E}_A[((1 + \epsilon_1)\boldsymbol{u}_{C(i)} + (1 + \epsilon_2)\boldsymbol{v}_{C(i)} + \boldsymbol{\xi})((1 + \epsilon_1)\boldsymbol{u}_{C(i)} + (1 + \epsilon_2)\boldsymbol{v}_{C(i)} + \boldsymbol{\xi})^\top]$$

$$= (1 + \sigma_1^2)\boldsymbol{u}_{C(i)}\boldsymbol{u}_{C(i)}^\top + \boldsymbol{v}_{C(i)}\boldsymbol{u}_{C(i)}^\top + \boldsymbol{u}_{C(i)}\boldsymbol{v}_{C(i)}^\top + (1 + \sigma_2^2)\boldsymbol{v}_{C(i)}\boldsymbol{v}_{C(i)}^\top$$

$$+ \frac{\phi^2}{d}(\boldsymbol{I}_d - \boldsymbol{u}_{C(i)}\boldsymbol{u}_{C(i)}^\top - \boldsymbol{v}_{C(i)}\boldsymbol{v}_{C(i)}^\top)$$

Therefore

$$\frac{1}{n}\sum_{i=1}^{n}\mathbb{E}_A[A(\boldsymbol{x}_i)A(\boldsymbol{x}_i)^\top] = \frac{1}{n}\sum_{i=1}^{n}(1+\sigma_1^2)\boldsymbol{u}_{C(i)}\boldsymbol{u}_{C(i)}^\top + \boldsymbol{u}_{C(i)}\boldsymbol{v}_{C(i)}^\top(1+\sigma_2^2)\boldsymbol{v}_{C(i)}\boldsymbol{v}_{C(i)}^\top + \boldsymbol{v}_{C(i)}\boldsymbol{u}_{C(i)}^\top$$

$$+ \frac{\phi^2}{d}(\boldsymbol{I}_d - \boldsymbol{u}_{C(i)}\boldsymbol{u}_{C(i)}^\top - \boldsymbol{v}_{C(i)}\boldsymbol{v}_{C(i)}^\top)$$

$$= \frac{1}{C}\left(\sum_{c=1}^{C}\alpha_1\boldsymbol{u}_c\boldsymbol{u}_c^\top + \boldsymbol{u}_c\boldsymbol{v}_c^\top + \boldsymbol{v}_c\boldsymbol{u}_c^\top + \alpha_2\boldsymbol{v}_c\boldsymbol{v}_c^\top\right) + \frac{\phi^2}{d}(\boldsymbol{I}_d - \sum_{c=1}^{C}\boldsymbol{u}_c\boldsymbol{u}_c^\top - \boldsymbol{v}_c\boldsymbol{v}_c^\top)$$

where we set $\alpha_1 = 1 + \sigma_1^2 + \frac{(C-1)\phi^2}{d}, \alpha_2 = 1 + \sigma_2^2 + \frac{(C-1)\phi^2}{d}$. Taking the inverse,

$$\left(\frac{1}{n}\sum_{i=1}^{n}\mathbb{E}_A[A(\boldsymbol{x}_i)A(\boldsymbol{x}_i)^\top]\right)^{-1} = \frac{C}{\alpha_1\alpha_2 - 1}\left(\sum_{c=1}^{C}\alpha_2\boldsymbol{u}_c\boldsymbol{u}_c^\top - \boldsymbol{u}_c\boldsymbol{v}_c^\top - \boldsymbol{v}_c\boldsymbol{u}_c^\top + \alpha_1\boldsymbol{v}_c\boldsymbol{v}_c^\top\right)$$

$$+ \frac{d}{\phi^2}(\boldsymbol{I}_d - \sum_{c=1}^{C}\boldsymbol{u}_c\boldsymbol{u}_c^\top - \boldsymbol{v}_c\boldsymbol{v}_c^\top)$$

Also, we have

$$\frac{1}{n}\sum_{i=1}^{n}\mathbb{E}_A[\boldsymbol{e}_i A(\boldsymbol{x}_i)^\top] = \frac{1}{n}\sum_{i=1}^{n}\mathbb{E}_A[\boldsymbol{e}_i((1+\epsilon_1)\boldsymbol{u}_{C(i)} + (1+\epsilon_2)\boldsymbol{v}_{C(i)} + \boldsymbol{\xi})^\top]$$

$$= \frac{1}{n}\sum_{i=1}^{n}\boldsymbol{e}_i(\boldsymbol{u}_{C(i)} + \boldsymbol{v}_{C(i)})^\top$$

Now using the previously calculated expressions,

$$\boldsymbol{W}\boldsymbol{u}_c = \boldsymbol{W}_H^\top\left(\frac{1}{n}\sum_{i=1}^{n}\mathbb{E}_A[\boldsymbol{e}_i A(\boldsymbol{x}_i)^\top]\right)\left(\frac{1}{n}\sum_{i=1}^{n}\mathbb{E}_A[A(\boldsymbol{x}_i)A(\boldsymbol{x}_i)^\top]\right)^{-1}\boldsymbol{u}_c$$

$$= \boldsymbol{W}_H^\top\left(\frac{1}{n}\sum_{i=1}^{n}\mathbb{E}_A[\boldsymbol{e}_i A(\boldsymbol{x}_i)^\top]\right)\left(\frac{C\alpha_2}{\alpha_1\alpha_2 - 1}\boldsymbol{u}_c - \frac{C}{\alpha_1\alpha_2 - 1}\boldsymbol{v}_c\right)$$

$$= \boldsymbol{W}_H^\top\left(\frac{1}{n}\sum_{C(i)=c}\left(\frac{C\alpha_2}{\alpha_1\alpha_2 - 1} - \frac{C}{\alpha_1\alpha_2 - 1}\right)\boldsymbol{e}_i\right)$$

$$= \frac{C(\alpha_2 - 1)}{n(\alpha_1\alpha_2 - 1)}\sum_{C(i)=c}\boldsymbol{W}_H^\top\boldsymbol{e}_i$$

$$= \frac{C(\sigma_2^2 + \frac{(C-1)\phi^2}{d})}{n(\alpha_1\alpha_2 - 1)}\sum_{C(i)=c}\boldsymbol{W}_H^\top\boldsymbol{e}_i$$

Similarly, we have

$$\boldsymbol{W}\boldsymbol{v}_c = \frac{C(\sigma_1^2 + \frac{(C-1)\phi^2}{d})}{n(\alpha_1\alpha_2 - 1)}\sum_{C(i)=c}\boldsymbol{W}_H^\top\boldsymbol{e}_i$$

It follows that

$$
\begin{aligned}
\frac{\|\boldsymbol{W}\boldsymbol{v}_c\|}{\|\boldsymbol{W}\boldsymbol{u}_c\|} &= \frac{\frac{C(\sigma_1^2 + \frac{(C-1)\phi^2}{d})}{n(\alpha_1\alpha_2-1)}}{\frac{C(\sigma_2^2 + \frac{(C-1)\phi^2}{d})}{n(\alpha_1\alpha_2-1)}} \\
&= \frac{\sigma_1^2 + \frac{(C-1)\phi^2}{d}}{\sigma_2^2 + \frac{(C-1)\phi^2}{d}}
\end{aligned}
$$

Using the fact that $C = o(d)$, this shows that $\frac{\|\boldsymbol{W}\boldsymbol{v}_c\|}{\|\boldsymbol{W}\boldsymbol{u}_c\|} = \frac{\sigma_1^2}{\sigma_2^2} + o(1)$, as desired. $\qquad\square$

For normalized diet, we prove the result via the following lemmas. First we define some notation.

Let $C(i) \in \mathcal{C}$ represent the concept associated with $\boldsymbol{x}_i$, and set $\boldsymbol{r}_c = \sum_{C(i)=c} \boldsymbol{W}_H^\top \boldsymbol{e}_i$. Also as shorthand we write

$$
\mathcal{L}_{diet-norm}^{(i)} = \frac{1}{2}\mathbb{E}_A[\|\boldsymbol{W}_H(norm(\boldsymbol{W}(A(\boldsymbol{x}_i)))) - \boldsymbol{e}_i\|^2]
$$

$$
\mathcal{L}_{diet-norm}^{MSE} = \frac{1}{n}\sum_{i=1}^{n} \mathcal{L}_{diet-norm}^{(i)}
$$

**Lemma B.5** (Useful facts). *In the assumed setting, the following hold*

1. *If $i \neq j$, then $(\boldsymbol{W}_H^\top \boldsymbol{e}_i)^\top(\boldsymbol{W}_H^\top \boldsymbol{e}_j) = 0$*

2. *If $C(i) = c$, then $\boldsymbol{r}_c^\top \boldsymbol{W}_H^\top \boldsymbol{e}_i = h_i^2$.*

*Proof.* Since $\boldsymbol{W}_H$ is an isometry by assumption, $\boldsymbol{W}_H^\top$ is a partial isometry. Since $\boldsymbol{e}_i \perp \boldsymbol{e}_j$, the first claim follows.

For the second claim, we calculate that

$$
\begin{aligned}
\boldsymbol{r}_c^\top \boldsymbol{W}_H^\top \boldsymbol{e}_i &= \sum_{C(j)=c} (\boldsymbol{W}_H^\top \boldsymbol{e}_j)^\top \boldsymbol{W}_H^\top \boldsymbol{e}_i \\
&= (\boldsymbol{W}_H^\top \boldsymbol{e}_i)^\top \boldsymbol{W}_H^\top \boldsymbol{e}_i \\
&= h_i^2
\end{aligned}
$$

$\square$

**Lemma B.6** (Step 1). *When training with $\mathcal{L}_{diet-norm}^{MSE}$, at every step in training $\boldsymbol{W}\boldsymbol{u}_c$ and $\boldsymbol{W}\boldsymbol{v}_c$ are parallel to $\boldsymbol{r}_c$, and $\boldsymbol{W}\boldsymbol{p} = 0$ for any $\boldsymbol{p}$ orthogonal to all the $\boldsymbol{u}_c$ and $\boldsymbol{v}_c$.*

*Proof.* We proceed by induction on the iteration of SGD.

The base case follows from the initialization $\boldsymbol{W} = 0$.

For the inductive step, we calculate the change due to the gradient descent update.

We first note that the inductive hypothesis implies the following useful fact: if $C(i) = c$ and $\boldsymbol{q} \in \mathbb{R}^d$ is orthogonal to $\boldsymbol{u}_c$ and $\boldsymbol{v}_c$, then $\boldsymbol{W}\boldsymbol{q} \in Span(\{\boldsymbol{r}_{c'} : c' \neq c\})$. In particular, by Lemma B.5, $\boldsymbol{W}\boldsymbol{q}$ and $\boldsymbol{W}_H^\top \boldsymbol{e}_i$ are orthogonal.

Now denoting $\boldsymbol{x}_i^A = A(\boldsymbol{x}_i)$, $\boldsymbol{z}_i^A = \boldsymbol{W}\boldsymbol{x}_i^A$, the gradient is

$$\frac{\partial \mathcal{L}_{diet-norm}^{(i)}}{\partial \boldsymbol{W}} = \mathbb{E}_A \left[ \frac{1}{\|\boldsymbol{z}_i^A\|} \left( \boldsymbol{I} - \frac{1}{\|\boldsymbol{z}_i^A\|^2} \boldsymbol{z}_i^A (\boldsymbol{z}_i^A)^\top \right) \boldsymbol{W}_H^\top (\boldsymbol{W}_H \boldsymbol{z}_i^A - \boldsymbol{e}_i)(\boldsymbol{x}_i^A)^\top \right]$$

$$= \mathbb{E}_A \left[ \frac{1}{\|\boldsymbol{z}_i^A\|} \left( \boldsymbol{I} - \frac{1}{\|\boldsymbol{z}_i^A\|^2} \boldsymbol{z}_i^A (\boldsymbol{z}_i^A)^\top \right) \boldsymbol{z}_i^A (\boldsymbol{x}_i^A)^\top - \frac{1}{\|\boldsymbol{z}_i^A\|} \left( \boldsymbol{I} - \frac{1}{\|\boldsymbol{z}_i^A\|^2} \boldsymbol{z}_i^A (\boldsymbol{z}_i^A)^\top \right) \boldsymbol{W}_H^\top \boldsymbol{e}_i (\boldsymbol{x}_i^A)^\top \right]$$

$$= -\mathbb{E}_A \left[ \frac{1}{\|\boldsymbol{z}_i^A\|} \left( \boldsymbol{I} - \frac{1}{\|\boldsymbol{z}_i^A\|^2} \boldsymbol{z}_i^A (\boldsymbol{z}_i^A)^\top \right) \boldsymbol{W}_H^\top \boldsymbol{e}_i (\boldsymbol{x}_i^A)^\top \right]$$

Thus

$$\frac{\partial \mathcal{L}_{diet-norm}^{(i)}}{\partial \boldsymbol{W}} \boldsymbol{u}_c = -\mathbb{E}_A \left[ \frac{(\boldsymbol{x}_i^A)^\top \boldsymbol{u}_c}{\|\boldsymbol{z}_i^A\|} \left( \boldsymbol{I} - \frac{1}{\|\boldsymbol{z}_i^A\|^2} \boldsymbol{z}_i^A (\boldsymbol{z}_i^A)^\top \right) \boldsymbol{W}_H^\top \boldsymbol{e}_i \right]$$

We now consider two cases. First assume $C(i) = c$. Writing $\boldsymbol{x}_i^A = (1+\epsilon_1)\boldsymbol{u}_c + (1+\epsilon_2)\boldsymbol{v}_c + \boldsymbol{\xi}$, and $\boldsymbol{W}((1+\epsilon_1)\boldsymbol{u}_c + (1+\epsilon_2)\boldsymbol{v}_c) = \alpha_c \boldsymbol{r}_c$

$$\frac{\partial \mathcal{L}_{diet-norm}^{(i)}}{\partial \boldsymbol{W}} \boldsymbol{u}_c = \mathbb{E}_A \left[ \frac{1+\epsilon_1}{\|\boldsymbol{z}_i^A\|} \left( \boldsymbol{I} - \frac{1}{\|\boldsymbol{z}_i^A\|^2} (\alpha_c \boldsymbol{r}_c + \boldsymbol{W}\boldsymbol{\xi})(\alpha_c \boldsymbol{r}_c + \boldsymbol{W}\boldsymbol{\xi})^\top \right) \boldsymbol{W}_H^\top \boldsymbol{e}_i \right]$$

Now by the symmetry of the noise distribution, we can replace $\boldsymbol{\xi}$ with $-\boldsymbol{\xi}$. By induction, $\boldsymbol{W}\boldsymbol{\xi}$ is orthogonal to $\boldsymbol{r}_c$, this does not change $\|\boldsymbol{z}_i\|$, so the above is equal to

$$= \mathbb{E}_A \left[ \frac{1+\epsilon_1}{\|\boldsymbol{z}_i^A\|} \left( \boldsymbol{I} - \frac{1}{2\|\boldsymbol{z}_i^A\|^2} ((\alpha_c \boldsymbol{r}_c + \boldsymbol{W}\boldsymbol{\xi})(\alpha_c \boldsymbol{r}_c + \boldsymbol{W}\boldsymbol{\xi})^\top + (\alpha_c \boldsymbol{r}_c - \boldsymbol{W}\boldsymbol{\xi})(\alpha_c \boldsymbol{r}_c - \boldsymbol{W}\boldsymbol{\xi})^\top) \right) \boldsymbol{W}_H^\top \boldsymbol{e}_i \right]$$

$$= \mathbb{E}_A \left[ \frac{1+\epsilon_1}{\|\boldsymbol{z}_i^A\|} \left( \boldsymbol{I} - \frac{1}{\|\boldsymbol{z}_i^A\|^2} (\alpha_c^2 \boldsymbol{r}_c \boldsymbol{r}_c^\top + (\boldsymbol{W}\boldsymbol{\xi})(\boldsymbol{W}\boldsymbol{\xi})^\top) \right) \boldsymbol{W}_H^\top \boldsymbol{e}_i \right]$$

Using the useful fact from above and Lemma B.5, this is equal to

$$= \mathbb{E}_A \left[ \frac{1+\epsilon_1}{\|\boldsymbol{z}_i^A\|} \boldsymbol{W}_H^\top \boldsymbol{e}_i - \frac{(1+\epsilon_1)\alpha_c^2 \boldsymbol{r}_c^\top \boldsymbol{W}_H^\top \boldsymbol{e}_i}{\|\boldsymbol{z}_i^A\|^3} \boldsymbol{r}_c \right]$$

$$= \mathbb{E}_A \left[ \frac{1+\epsilon_1}{\|\boldsymbol{z}_i^A\|} \boldsymbol{W}_H^\top \boldsymbol{e}_i - \frac{(1+\epsilon_1)\alpha_c^2 h_c^2}{\|\boldsymbol{z}_i^A\|^3} \boldsymbol{r}_c \right]$$

Now suppose $C(i) = c' \neq C$. A similar calculation shows that

$$\frac{\partial \mathcal{L}_{diet-norm}^{(i)}}{\partial \boldsymbol{W}} \boldsymbol{u}_c = \mathbb{E}_A \left[ \frac{\boldsymbol{\xi}^\top \boldsymbol{u}_c}{\|\boldsymbol{z}_i^A\|} \left( \boldsymbol{I} - \frac{1}{\|\boldsymbol{z}_i^A\|^2} (\alpha_{c'} \boldsymbol{r}_{c'} + \boldsymbol{W}\boldsymbol{\xi})(\alpha_{c'} \boldsymbol{r}_{c'} + \boldsymbol{W}\boldsymbol{\xi})^\top \right) \boldsymbol{W}_H^\top \boldsymbol{e}_i \right]$$

Again using the symmetry of the noise and the useful fact, this is equal to

$$= \frac{1}{2} \mathbb{E}_A \left[ \frac{\boldsymbol{\xi}^\top \boldsymbol{u}_c}{\|\boldsymbol{z}_i^A\|} (\boldsymbol{I} - \frac{1}{\|\boldsymbol{z}_i^A\|^2} (\alpha_{c'} \boldsymbol{r}_{c'} + \boldsymbol{W}\boldsymbol{\xi})(\alpha_{c'} \boldsymbol{r}_{c'} + \boldsymbol{W}\boldsymbol{\xi})^\top) \boldsymbol{W}_H^\top \boldsymbol{e}_i \right.$$

$$\left. + \frac{-\boldsymbol{\xi}^\top \boldsymbol{u}_c}{\|\boldsymbol{z}_i^A\|} (\boldsymbol{I} - \frac{1}{\|\boldsymbol{z}_i^A\|^2} (\alpha_{c'} \boldsymbol{r}_{c'} - \boldsymbol{W}\boldsymbol{\xi})(\alpha_{c'} \boldsymbol{r}_{c'} - \boldsymbol{W}\boldsymbol{\xi})^\top) \boldsymbol{W}_H^\top \boldsymbol{e}_i \right]$$

$$= -\mathbb{E}_A \left[ \frac{\boldsymbol{\xi}^\top \boldsymbol{u}_c}{\|\boldsymbol{z}_i^A\|^3} \left( \alpha_{c'} \boldsymbol{r}_{c'} (\boldsymbol{W}\boldsymbol{\xi})^\top + (\boldsymbol{W}\boldsymbol{\xi})(\alpha_{c'} \boldsymbol{r}_{c'})^\top \right) \boldsymbol{W}_H^\top \boldsymbol{e}_i \right]$$

$$= -\mathbb{E}_A \left[ \frac{\alpha_{c'} (\boldsymbol{\xi}^\top \boldsymbol{u}_c)(\boldsymbol{r}_{c'}^\top \boldsymbol{W}_H^\top \boldsymbol{e}_i)}{\|\boldsymbol{z}_i^A\|^3} \boldsymbol{W}\boldsymbol{\xi} \right]$$

$$= -\mathbb{E}_A \left[ \frac{\alpha_{c'} h_{c'}^2 (\boldsymbol{\xi}^\top \boldsymbol{u}_c)}{\|\boldsymbol{z}_i^A\|^3} \boldsymbol{W}\boldsymbol{\xi} \right]$$

Now isolating the component of $\boldsymbol{\xi}$ along $\boldsymbol{u}_c$, write $\boldsymbol{\xi} = \xi_c \boldsymbol{u}_c + \boldsymbol{\xi}'$. Again by the symmetry of the noise, we can consider replacing $\boldsymbol{\xi}'$ with $-\boldsymbol{\xi}$, so

$$-\mathbb{E}_A \left[ \frac{\alpha_{c'} h_{c'}^2 (\boldsymbol{\xi}^\top \boldsymbol{u}_c)}{\|\boldsymbol{z}_i\|^3} \boldsymbol{W} \boldsymbol{\xi} \right] = -\mathbb{E}_A \left[ \frac{\alpha_{c'} h_{c'}^2 \xi_c}{2\|\boldsymbol{z}_i\|^3} \boldsymbol{W} (\xi_c \boldsymbol{u}_c + \boldsymbol{\xi}' + \xi_c \boldsymbol{u}_c - \boldsymbol{\xi}') \right]$$

$$= -\mathbb{E}_A \left[ \frac{\alpha_{c'} h_{c'}^2 \xi_c^2}{\|\boldsymbol{z}_i\|^3} \boldsymbol{W} \boldsymbol{u}_c \right]$$

Combining all these results, we have

$$\frac{\partial \mathcal{L}_{diet-norm}^{MSE}}{\partial \boldsymbol{W}} \boldsymbol{u}_c = \frac{1}{n} \sum_{i=1}^{n} \frac{\partial \mathcal{L}_{diet-norm}^{(i)}}{\partial \boldsymbol{W}}$$

$$= -\frac{1}{n} \sum_{C(i)=c} \mathbb{E}_A \left[ \frac{1 + \epsilon_1^{(i)}}{\|\boldsymbol{z}_i^A\|} \boldsymbol{W}_H^\top \boldsymbol{e}_i - \frac{(1 + \epsilon_1^{(i)})(\alpha_c^{(i)})^2 h_c^2}{\|\boldsymbol{z}_i^A\|^3} \boldsymbol{r}_c \right]$$

$$+ \frac{1}{n} \sum_{C(i)=c'\neq c} \mathbb{E}_A \left[ \frac{\alpha_{c'}^{(i)} (\xi_c^{(i)})^2 h_{c'}^2}{\|\boldsymbol{z}_i^A\|^3} \boldsymbol{W} \boldsymbol{u}_c \right]$$

$$= -\mathbb{E}_A \left[ \frac{1 + \epsilon_1}{\|\boldsymbol{z}^A\|} \right] \boldsymbol{r}_c + \frac{1}{n} \sum_{C(i)=c} \mathbb{E}_A \left[ \frac{(1 + \epsilon_1^{(i)})(\alpha_c^{(i)})^2 h_c^2}{\|\boldsymbol{z}_i^A\|^3} \right] \boldsymbol{r}_c$$

$$+ \frac{1}{n} \sum_{C(i)=c'\neq c} \mathbb{E}_A \left[ \frac{\alpha_{c'}^{(i)} (\xi_c^{(i)})^2 h_{c'}^2}{\|\boldsymbol{z}_i^A\|^3} \right] \boldsymbol{W} \boldsymbol{u}_c$$

By the inductive hypothesis $\boldsymbol{W} \boldsymbol{u}_c$ is parallel to $\boldsymbol{r}_c$, so the change from the gradient update is parallel to $\boldsymbol{r}_c$.

The same argument shows that $\boldsymbol{W} \boldsymbol{v}_c$ is parallel to $\boldsymbol{r}_c$.

Now consider any $\boldsymbol{p}$ orthogonal to all the $\boldsymbol{v}_i$ and $\boldsymbol{u}_i$. We calculate that

$$\frac{\partial \mathcal{L}_{diet-norm}^{MSE}}{\partial \boldsymbol{W}} \boldsymbol{p} = -\frac{1}{n} \sum_{i=1}^{n} \mathbb{E}_A \left[ \frac{(\boldsymbol{x}_i^A)^\top \boldsymbol{p}}{\|\boldsymbol{z}_i^A\|} \left( \boldsymbol{I} - \frac{1}{\|\boldsymbol{z}_i\|^2} \boldsymbol{z}_i^A (\boldsymbol{z}_i^A)^\top \right) \boldsymbol{W}_H^\top \boldsymbol{e}_i \right]$$

Decomposing $\boldsymbol{x} = \beta \boldsymbol{p} + \gamma$, by the symmetry of the noise we can replace $\beta$ with $-\beta$. Since $\boldsymbol{W} \boldsymbol{p} = \boldsymbol{0}$ by induction $\boldsymbol{z}_i$ does not change, so we have

$$\frac{\partial \mathcal{L}_{diet-norm}^{MSE}}{\partial \boldsymbol{W}} \boldsymbol{p} = -\frac{1}{2n} \sum_{i=1}^{n} \mathbb{E}_A \left[ \frac{(\boldsymbol{x}_i^A)^\top \boldsymbol{p} - (\boldsymbol{x}_i^A)^\top \boldsymbol{p}}{\|\boldsymbol{z}_i^A\|} \left( \boldsymbol{I} - \frac{1}{\|\boldsymbol{z}_i\|^2} \boldsymbol{z}_i^A (\boldsymbol{z}_i^A)^\top \right) \boldsymbol{W}_H^\top \boldsymbol{e}_i \right]$$

$$= \boldsymbol{0}$$

Thus the change from the gradient update is $\boldsymbol{0}$,

This completes the induction. $\qquad \square$

**Lemma B.7** (Step 2). *Assume that we train to convergence using $\mathcal{L}_{diet-norm}^{MSE}$. Then $\boldsymbol{r}_c^\top \boldsymbol{W} \boldsymbol{u}_c, \boldsymbol{r}_c^\top \boldsymbol{W} \boldsymbol{v}_c \neq 0$.*

*Proof.* Using the gradient calculation from the proof of step 1:

$$\boldsymbol{r}_c^\top \frac{\partial \mathcal{L}_{diet-norm}^{MSE}}{\partial \boldsymbol{W}} \boldsymbol{u}_c = -\mathbb{E}_A \left[ \frac{1+\epsilon_1}{\|\boldsymbol{z}^A\|} \right] \|\boldsymbol{r}_c\|^2 + \frac{1}{n} \sum_{C(i)=c} \mathbb{E}_A \left[ \frac{(1+\epsilon_1^{(i)})(\alpha_c^{(i)})^2 h_c^2}{\|\boldsymbol{z}_i^A\|^3} \right] \|\boldsymbol{r}_c\|^2$$

$$+ \frac{1}{n} \sum_{C(i)=c'\neq c} \mathbb{E}_A \left[ \frac{\alpha_{c'}^{(i)}(\xi_c^{(i)})^2 h_{c'}^2}{\|\boldsymbol{z}_i^A\|^3} \right] \boldsymbol{r}_c^\top \boldsymbol{W} \boldsymbol{u}_c$$

$$= -\mathbb{E}_A \left[ \frac{1+\epsilon_1}{\|\boldsymbol{z}^A\|} \right] \|\boldsymbol{r}_c\|^2 + \frac{1}{n} \sum_{C(i)=c} \mathbb{E}_A \left[ \frac{(1+\epsilon_1^{(i)})(\alpha_c^{(i)})^2 h_c^2}{\|\boldsymbol{z}_i^A\|^3} \right] \|\boldsymbol{r}_c\|^2$$

Recall from Lemma B.5 that $h_c^2 \leq 1$. Also note that by definition $(\alpha_c^{(i)})^2 \leq \|\boldsymbol{z}_i\|^2$. Hence

$$\boldsymbol{r}_c^\top \frac{\partial \mathcal{L}_{diet-norm}^{MSE}}{\partial \boldsymbol{W}} \boldsymbol{u}_c = -\mathbb{E}_A \left[ \frac{1+\epsilon_1}{\|\boldsymbol{z}^A\|} \right] + \frac{1}{n} \sum_{C(i)=c} \mathbb{E}_A \left[ \frac{(1+\epsilon_1^{(i)})(\alpha_c^{(i)})^2 h_c^2}{\|\boldsymbol{z}_i^A\|^3} \right] < 0$$

This implies that $\boldsymbol{r}_c^\top \frac{\partial \mathcal{L}_{diet-norm}^{MSE}}{\partial \boldsymbol{W}} \boldsymbol{u}_c < 0$. This contradicts the fact that we have converged to a point where $\frac{\partial \mathcal{L}_{diet-norm}^{MSE}}{\partial \boldsymbol{W}} = \boldsymbol{0}$. □

**Lemma B.8** (Proof of Theorem for Normalized Diet). *Assume that we train to convergence using $\mathcal{L}_{diet}^{norm}$. Then*

$$\frac{1-\nu_1}{1+\nu_2} \leq \frac{\|\boldsymbol{W}\boldsymbol{v}_c\|}{\|\boldsymbol{W}\boldsymbol{u}_c\|} \leq \frac{1+\nu_1}{1-\nu_2}$$

*Proof.* From the previous steps, there exists $a_1, \ldots, a_C, b_1, \ldots, b_C \neq 0$ such that $\boldsymbol{W}\boldsymbol{u}_c = a_c \boldsymbol{r}_c$ and $\boldsymbol{W}\boldsymbol{v}_c = b_c \boldsymbol{r}_c$. Therefore, for a given example $\boldsymbol{x}_i$ with $C(i) = c$, the distribution $\boldsymbol{W}(A(\boldsymbol{x}_i))$ over choice of augmentation $A$ takes the form

$$(a_c + a_c\epsilon_1 + b_c + b_c\epsilon_2)\boldsymbol{r}_c + \sum_{c'\neq c} \frac{\phi^2}{d}\sqrt{a_{c'}^2 + b_{c'}^2}\xi_{c'}\boldsymbol{r}_{c'} \tag{17}$$

where $\epsilon_1 \sim \mathcal{G}_1, \epsilon_2 \sim \mathcal{G}_2$, and $\xi_{c'} \sim \mathcal{N}(0,1)$ for each $c'$. To ease notation, set $\kappa_c = a_c + a_c\epsilon_1 + b_c + b_c\epsilon_2$ and $\lambda_c = \frac{\phi^2}{d}\sqrt{a_c^2 + b_c^2}\xi_c$. Note that since the $\boldsymbol{r}_c$ are orthogonal, $\|\boldsymbol{W}(A(\boldsymbol{x}_i))\|$ follows the distribution

$$\sqrt{\kappa_c^2\|\boldsymbol{r}_c\|^2 + \sum_{c'\neq c} \lambda_{c'}^2\|\boldsymbol{r}_{c'}\|^2}$$

We can now treat the loss as a multivariate function in $a_1, \ldots, a_C, b_1, \ldots, b_C$. Suppose we vary $a_c$ and $b_c$ such that $a_c da_c + b_c db_c = 0$. It suffices to calculate the directional derivative induced by this variation and show that it cannot be zero if $|\frac{b}{a}| > \frac{1+\nu_1}{1-\nu_2}$ or $|\frac{b}{a}| < \frac{1-\nu_1}{1+\nu_2}$.

The loss term due to an example $\boldsymbol{x}_i$ is

$$\mathcal{L}_{diet-norm}^{(i)} = \frac{1}{2}\mathbb{E}_A[\|\boldsymbol{W}_H(norm(\boldsymbol{W}(A(\boldsymbol{x}_i)))) - \boldsymbol{e}_i\|^2]$$

$$= \frac{1}{2}\mathbb{E}\left[\left\|\frac{\boldsymbol{W}_H(\kappa_{C(i)}\boldsymbol{r}_{C(i)} + \sum_{c'\neq C(i)}\lambda_{c'}\boldsymbol{r}_{c'})}{\sqrt{\kappa_{C(i)}^2\|\boldsymbol{r}_{C(i)}\|^2 + \sum_{c'\neq C(i)}\lambda_{c'}^2\|\boldsymbol{r}_{c'}\|^2}} - \boldsymbol{e}_i\right\|^2\right]$$

$$= 1 - \mathbb{E}\left[\frac{\boldsymbol{e}_i^\top \boldsymbol{W}_H(\kappa_{C(i)}\boldsymbol{r}_{C(i)} + \sum_{c'\neq C(i)}\lambda_{c'}\boldsymbol{r}_{c'})}{\sqrt{\kappa_{C(i)}^2\|\boldsymbol{r}_{C(i)}\|^2 + \sum_{c'\neq C(i)}\lambda_{c'}^2\|\boldsymbol{r}_{c'}\|^2}}\right]$$

$$= 1 - \mathbb{E}\left[\frac{\kappa_{C(i)}\boldsymbol{e}_i^\top \boldsymbol{W}_H\boldsymbol{r}_{C(i)}}{\sqrt{\kappa_{C(i)}^2\|\boldsymbol{r}_{C(i)}\|^2 + \sum_{c'\neq C(i)}\lambda_{c'}^2\|\boldsymbol{r}_{c'}\|^2}}\right]$$

$$= 1 - \mathbb{E}\left[\frac{\kappa_{C(i)}h_c^2}{\sqrt{\kappa_{C(i)}^2\|\boldsymbol{r}_{C(i)}\|^2 + \sum_{c'\neq C(i)}\lambda_{c'}^2\|\boldsymbol{r}_{c'}\|^2}}\right]$$

Observe that by construction $d(a_c^2 + b_c^2) = 0$, which implies $d\lambda_c = 0$. Thus if $C(i) \neq c$, the change in the loss $\mathcal{L}_{diet-norm}^{(i)}$ is zero.

On the other hand, if $C(i) = c$, we now calculate the derivatives

$$\frac{\partial}{\partial a_c}\mathcal{L}_{diet-norm}^{(i)} = \frac{\partial}{\partial a_c}\mathbb{E}_A[\|\boldsymbol{W}_H(norm(\boldsymbol{W}(A(\boldsymbol{x}_i)))) - \boldsymbol{e}_i\|^2]$$

$$= -\mathbb{E}\left[\frac{h_c^2 \sum_{c'\neq c}\lambda_{c'}^2\|\boldsymbol{r}_{c'}\|^2}{(\kappa_c^2\|\boldsymbol{r}_c\|^2 + \sum_{c'\neq c}\lambda_{c'}^2\|\boldsymbol{r}_{c'}\|^2)^{\frac{3}{2}}}(1 + \epsilon_1)\right]$$

$$\frac{\partial}{\partial b_c}\mathcal{L}_{diet-norm}^{(i)} = -\mathbb{E}\left[\frac{h_c^2 \sum_{c'\neq c}\lambda_{c'}^2\|\boldsymbol{r}_{c'}\|^2}{(\kappa_c^2\|\boldsymbol{r}_c\|^2 + \sum_{c'\neq c}\lambda_{c'}^2\|\boldsymbol{r}_{c'}\|^2)^{\frac{3}{2}}}(1 + \epsilon_2)\right]$$

Hence

$$d\mathcal{L}_{diet-norm}^{MSE} = \sum_{C(i)=c}\frac{\partial \mathcal{L}_{diet-norm}^{(i)}}{\partial a_c}da_c + \frac{\partial \mathcal{L}_{diet-norm}^{(i)}}{\partial b_c}db_c$$

$$= \sum_{C(i)=c} -\mathbb{E}\left[\frac{h_c^2 \sum_{c'\neq c}\lambda_{c'}^2\|\boldsymbol{r}_{c'}\|^2}{(\kappa_c^2\|\boldsymbol{r}_c\|^2 + \sum_{c'\neq c}\lambda_{c'}^2\|\boldsymbol{r}_{c'}\|^2)^{\frac{3}{2}}}((1 + \epsilon_1)da_c + (1 + \epsilon_2)db_c)\right]$$

First consider the case that $\frac{a}{b} > 0$. Suppose for the sake of contradiction $|\frac{a}{b}| > \frac{1+\nu_1}{1-\nu_2}$. Then

$$0 > 1 + \nu_1 - \frac{a}{b} + \frac{a}{b}\nu_2$$

$$> 1 + \epsilon_1 - \frac{a}{b} + \frac{a}{b}(-\epsilon_2)$$

$$= (1 + \epsilon_1) - \frac{a}{b}(1 + \epsilon_2)$$

In the case that $\frac{a}{b} < 0$

$$0 > 1 + \nu_1 - \left|\frac{a}{b}\right| + \left|\frac{a}{b}\right|\nu_2$$

$$> 1 - \epsilon_1 + \frac{a}{b} - \frac{a}{b}\epsilon_2$$

$$= 2 - ((1 + \epsilon_1) - \frac{a}{b}(1 + \epsilon_2))$$

$$(1 + \epsilon_1) - \frac{a}{b}(1 + \epsilon_2) > 2$$

Either way $(1 + \epsilon_1) - \frac{a}{b}(1 + \epsilon_2)$ is strictly positive or strictly negative. Now write

$$(1 + \epsilon_1)da_c + (1 + \epsilon_2)db_c = \left((1 + \epsilon_1) - \frac{a}{b}(1 + \epsilon_2)\right)da_c$$

Combined with the fact that $\frac{h_c^2 \sum_{c' \neq c} \lambda_{c'}^2 \|\boldsymbol{r}_{c'}\|^2}{(\kappa_c^2 \|\boldsymbol{r}_c\|^2 + \sum_{c' \neq c} \lambda_{c'}^2 \|\boldsymbol{r}_{c'}\|^2)^{\frac{3}{2}}}$ is always nonnegative and not always zero, it follows that $d\mathcal{L}_{diet-norm}^{MSE} \neq 0$, contradicting the fact that we have converged to a local minima.

The same argument shows that $\frac{b}{a} \leq \frac{1+\nu_2}{1-\nu_1}$, giving the lower bound. $\qquad\square$

## C  EXPERIMENTAL SETUP

Unless otherwise specified, we use the following setup, which aligns with that in Balestriero (2023).

- batch size of 256
- training schedule of 5000 epochs
- cross entropy loss with label smoothing of $0.8$
- ADAM-W optimizer with learning rate $0.001$, weight decay $0.05$.
- cosine learning rate scheduler
- model consists of a base model, projection head, and classifier head; we refer to the model by the base model architecture (e.g. Resnet-18 or Resnet-50); the projection head is a 3 layer ReLU MLP; classifier head is a linear layer without bias.
- representations are normalized before being passed to the classifier head
- augmentations include random resized crop with scale in (0.08, 1.0), random horizontal flip, random color jitter (brightness = 0.4, contrast = 0.4, saturation = 0.4, hue = 0.1), and random grayscale

For ResNet models, we remove the last linear layer. In addition, on CIFAR-10 and CIFAR-100, we modify the first convolution layer by reducing the kernel size from 7×7 to 3 × 3 and the stride from 2 to 1; the max pooling layer following it is removed.

### C.1  SSL METHODS

Pretrained models for all SSL methods are obtained using the solo-learn library (da Costa et al., 2022). We use the batch size and augmentations as specified in the previous section, and change the precision to 32-bit for consistency. All other hyperparameters are left unchanged.

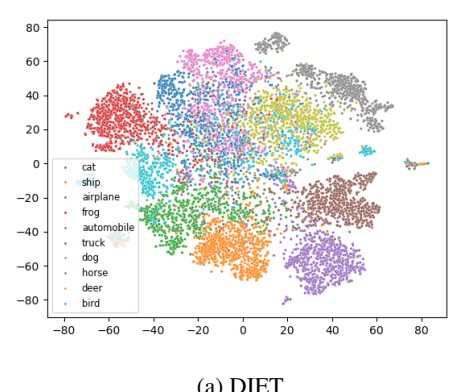

(a) DIET

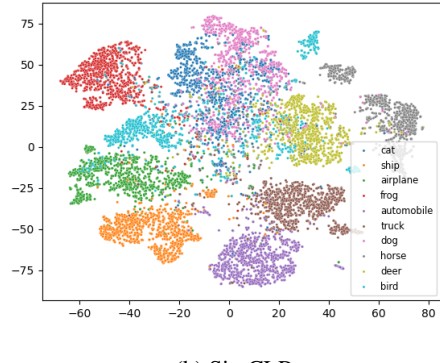

(b) SimCLR

Figure 2: TSNE of embeddings produced by DIET and SimCLR on CIFAR-10 using ResNet-50.

## C.2 TOY DATASET

We instantiate the scenario from Section 5.1 with a more realistic training setup:

- we make the classifier head $\boldsymbol{W}_H$ trainable from random initialization.
- instead of taking the expectation over all augmentations, we sample a single random augmentation of the input at each step.

We also choose $\mathcal{G}_1, \mathcal{G}_2$ to follow normal distributions. We set $C = 4, d = 16, m = 4, n = 32, \sigma = 0.01, \tau = 0.1, \phi = 0.001$. We train for 5000 steps using the Adam optimizer with learning rate $0.1$ and cosine learning rate schedule. We reset the state of the Adam optimizer after the first step to eliminate the effect of gradient blowup from normalizing zero vectors, see Appendix A.3 for details.

## C.3 SYNTHETIC DATASET

For the synthetic dataset described in Section 6.1.1, we modify the first convolutional layer of the ResNet model to take 4 input channels instead of 3. For MNIST augmentations, we replace random horizontal flip and random grayscale with gaussian blur. We also modify the random cropping to keep at least 0.75 of the area of the original image. We train for 500 epochs. All other hyperparameters are set as described above.

## D COMPARISON BETWEEN DIET AND CL

In Figure 2, we compare t-SNE visualizations (van der Maaten & Hinton, 2008) of test embeddings produced by S-DIET and SimCLR (Chen et al., 2020) on CIFAR-10 with ResNet-50. We observe that the high level structure of the embeddings is remarkably simlar for both methods.

## E PSEUDOCODE FOR S-DIET

```
"""
Uppercase variables stored on disk
Lowercase variables stored in memory

X: train data
```

```python
H: classifier head
M: first moment for classifier head
V: second moment for classifier head

indices: indices for the current batch
"""
def train_step(X, H, M, V, indices, model, criterion, optimizer):
    # Load data, head weights, and head optimizer state into memory
    inputs, head, optimizer_m, optimizer_v = X[indices], H[indices], M[indices], V[indices]
    labels = [0, 1, ..., len(indices)-1]

    # Forward and backward pass
    outputs = head(model(inputs))
    loss = criterion(outputs, labels)
    optimizer.zero_grad()
    loss.backward()
    optimizer.step()
    head, m, v = perform_multistep_adamw_head_update(head, m, v)

    # Save head weights and head optimizer state
    # Done asynchronously
    H[indices], M[indices], V[indices] = head, m, v

def perform_multistep_adamw_head_update(head, m, v):
    g = head.grad

    # first step
    head = (1 - lr * weight_decay) * head
    m = beta1 * m + (1 - beta1) * g
    v = beta2 * v + (1 - beta2) * g * g
    head = head - lr * m / (sqrt(v) + eps)

    # all other steps
    mu = beta1 / sqrt(beta2)
    alpha1 = (1 - lr * weight_decay) ** (t - 1)
    alpha2 = (alpha1 * lr * mu - lr * (mu ** t)) / (1 - lr * weight_decay - mu)

    head = alpha1 * head - alpha2 * m / (sqrt(v) + eps)
    m = (beta1 ** (t - 1)) * m
    v = (beta2 ** (t - 1)) * v
```

Listing 1: Pseudocode for a S-DIET training step

