# OpenReview forum: "Memory-Efficient Self-Supervised Contrastive Learning with a Supervised Loss"
_ICLR.cc/2025/Conference — Submitted to ICLR 2025_

### Official Review · Reviewer_cXwx · 2024-10-29

**Soundness:** 2
**Presentation:** 2
**Contribution:** 2
**Rating:** 5
**Confidence:** 4

**Summary:**

This study provides a theoretical analysis of DIET, a recently proposed supervised approach for self-supervised learning. DIET, as a menthod of CL, labels each example by its datum index and employs a supervised loss for training. This work obtains several conclusions, including (i) for linear encoders, DIET with MSE loss is equivalent to spectral contrastive loss; (ii) DIET tends to learn features with less noise but may not capture all relevant aspects of the training data; (iii) feature normalization can help mitigate this issue, while incorporating a projection head can further enhance performance. This work further introduces SCALED-DIET (S-DIET) to improve the model's linear probe accuracy.

**Strengths:**

- This paper explore the limitation of DIET, an important method of CL, and obtain several conclusions: (i) for linear encoders, DIET with MSE loss is equivalent to spectral contrastive loss; (ii) DIET tends to learn features with less noise but may not capture all relevant aspects of the training data; (iii) feature normalization can help mitigate this issue, while incorporating a projection head can further enhance performance.

- This work further introduces SCALED-DIET (S-DIET) to improve the model's linear probe accuracy, i.e., use batch cross entropy and the multistep update formula for AdamW.

- Some experiments demonstrate the effectiveness of the proposed S-DIET.

**Weaknesses:**

- The motivation behind this paper is unclear. According to P2 in the Introduction, DIET's advantage lies in its ability to mitigate CL's reliance on large datasets, while requiring a smaller parameter dimension to balance with sample size. The claim that smaller encoder dimensions are a key drawback for handling large data is not well justified. Furthermore, the authors state, "not clear whether DIET can capture the pairwise similarities between views...SSL," yet DIET, as a CL algorithm utilizing supervised loss, does not explicitly depend on pairwise similarities; this is merely an implementation choice rather than a fundamental mechanism. I fail to see how this motivation strongly connects DIET with contrastive loss, is pairwise similarity closely related to memory? While exploring CL from an efficiency perspective do be valuable, a thorough reading of the paper reveals a lack of such information. Perhaps I overlooked some details, and I hope the authors can clarify their insights.

- The key conclusions in this work need further explanation in relation to the core ides, i.e., MEMORY-EFFICIENT. For instance, the relationships between memory and features, encoder parameter dimensions, and projection heads are not adequately described, leading to fragmented conclusions.

- The choice to study DIET is justified by its independence from large training data, but there are other CL algorithms that also do not rely on large datasets, such as few-shot SSL. Additionally, DIET requires labeled information. Can the analyses in this work be applied to these other methods? If so, what differentiates this work? A broader exploration of algorithms and their mechanisms might enhance the reliability of this study.

- A code that can reflect the idea of ​​algorithm implementation is encouraged, since it is currently only described in 6 lines. At the same time, the introduction of these modules will increase the computational overhead, and related experiments are also necessary, after all, the focus is on memory.

- (Minor) The paper's template appears to differ from the one provided on the official website, such as in the line numbering. Please consider making further corrections.

**I would be happy to reconsider my score if these concerns can be addressed.**

**Questions:**

Please see **Weaknesses**.

---

> ### Author Response · Authors · 2024-11-21
>
> We thank the reviewer for their positive feedback about the theoretical and empirical contributions of our work and for their comments. We refer the reviewer to our general comment [(link)](https://openreview.net/forum?id=4NgxI6Z74n&noteId=Buw6JHWcGS) for a broader explanation of our work, but we hope to address the reviewer’s specific questions below:
>
> 1. To clarify, the motivation for this work is not to address the reliance on large datasets or the drawback of smaller encoders. The main motivation for the paper is that the paradigm of pairwise losses that currently dominate the SSL landscape suffers from practical difficulties—specifically regarding memory: (1) pairwise losses require maintaining multiple views of each example, (2) CL requires large batch sizes, and thus large GPU memory requirements. Using pairwise similarities is indeed a design choice, but we showed that this is not required and much simpler methods can learn high-quality representations. Specifically, we show that theoretically the spectral contrastive loss and DIET with MSE loss are equivalent for linear models, and empirically S-DIET can match the performance of SSL methods. This demonstrates that supervised methods are expressive enough to be used in place of pairwise losses while avoiding the difficulties with the latter, such as high memory usage. To our knowledge, both our results are novel in the literature. We hope this clarifies the motivation for this work and have clarified this in the updated version.
>
> 2. We proposed a memory-efficient method for representation learning (S-DIET). Feature normalization and projection head are common practices in CL literature that we applied to our memory-efficient DIET to further boost its performance, and confirm that S-DIET achieves a comparable or superior performance to CL. While the purpose of these additions is not to reduce memory usage, they are essential to obtain optimal performance. The memory efficiency of our method comes from the fact that we replaced complicated pairwise losses with a simple supervised loss and that we no longer need to maintain a massive classifier head in memory. We will clarify this in the revised version.
>
> 3. We note that DIET does not require labels, the labels used in the DIET loss are pseudo-labels that are simply the datum index. Thus the setting for few shot SSL methods is different from the fully self-supervised setting of DIET, and it is not immediate how to apply the existing analysis in this new setting. Seeing whether the analysis can be extended to these methods would be interesting but is beyond the scope of the current work. In the existing setting, our theoretical and empirical results are, to our knowledge, the first of their kind.
>
> 4. We thank the reviewer for the suggestion. We have included the following pseudocode for a single training step in the revised manuscript (page 27).
> ```
> """
> Uppercase variables stored on disk
> Lowercase variables stored in memory
>
> X: train data
> H: classifier head
> M: first moment for classifier head
> V: second moment for classifier head
>
> indices: indices for the current batch
> """
> def train_step(X, H, M, V, indices, model, criterion, optimizer):
>     # Load data, head weights, and head optimizer state into memory
>     inputs, head, optimizer_m, optimizer_v = X[indices], H[indices], M[indices], V[indices]
>     labels = [1, 2, ..., len(indices)]
>
>     # Forward and backward pass
>     outputs = head(model(inputs))
>     loss = criterion(outputs, labels)
>     optimizer.zero_grad()
>     loss.backward()
>     optimizer.step()
>     head, m, v = perform_multistep_adamw_head_update(head, m, v)
>
>     # Save head weights and head optimizer state
>     # Done asynchronously
>     H[indices], M[indices], V[indices] = head, m, v
>
>
> def perform_multistep_adamw_head_update(head, m, v):
>     g = head.grad
>
>     # first step
>     head = (1 - lr * weight_decay) * head
>     m = beta1 * m + (1 - beta1) * g
>     v = beta2 * v + (1 - beta2) * g * g
>     head = head - lr * m / (sqrt(v) + eps)
>
>     # all other steps
>     mu = beta1 / sqrt(beta2)
>     alpha1 = (1 - lr * weight_decay) ** (t - 1)
>     alpha2 = (alpha1 * lr * mu - lr * (mu ** t)) / (1 - lr * weight_decay - mu)
>
>     head = alpha1 * head - alpha2 * m / (sqrt(v) + eps)
>     m = (beta1 ** (t - 1)) * m
>     v = (beta2 ** (t - 1)) * v
>
> ```
> 5. We thank the reviewer for pointing this out, we have updated the formatting of the paper.

---

> > ### Author Response · Authors · 2024-11-24
> >
> > We hope our rebuttal has addressed your concerns. As we are getting close to the end of the discussion session, we would like to reach out and see if you had a chance to read our rebuttal and if there is anything else we can further clarify? We are looking forward to your response.

---

> ### Comment · Reviewer_cXwx · 2024-11-25
> **Official Comment by Reviewer cXwx**
>
> Thank you for the response, but my concerns remain unresolved.
>
> First, the authors claim that their motivation stems from the "currently dominate the SSL landscape suffers from practical difficulties," and that "DIET eliminates the reliance on pairwise similarities, thus reducing memory requirements." However, the pseudo-label-based supervised information in DIET similarly relies on approximate view invariance, which is fundamentally consistent. The equivalence of contrastive loss and MSE loss under this approximate view invariance has been widely acknowledged in the SSL community (e.g., [1-4]). Moreover, I reviewed the references cited by the authors in L37-38, but found no theoretical support for the connection between pairwise similarities and high memory usage, hoping the authors can clarify this further.
> [1] Representation learning with contrastive predictive coding; [2] What makes for good views for contrastive learning?; [3] Learning deep representations by mutual information estimation and maximization; [4] contrastive learning can find an optimal basis for approximately view-invariant functions.
>
> Second, regarding the methodology, the authors mention applying CL constraints on the projection head back to DIET, but the motivation behind this is unclear, especially as the mentioned "the purpose of these additions is not to reduce memory usage". If the method only aims to increase the accuracy points without considering the connection with motivation, it may not be reliable enough since the proposed issues still remain unsolved. Furthermore, the authors emphasize that the core of their algorithm is to "replace complicated pairwise losses with a simple supervised loss," yet this seems to overlap with DIET and prior works such as [1-4].
>
> As I detailed in W1 & W2, my primary concern lies in the connection between the work's motivation, specific implementation, and conclusions. Also mentioned in the comments that if this can be adequately addressed, I would be happy to adjust my score, but the current rebuttal and revision may require further clarification.

---

### Official Review · Reviewer_umd8 · 2024-10-31

**Soundness:** 2
**Presentation:** 3
**Contribution:** 2
**Rating:** 3
**Confidence:** 4

**Summary:**

The manuscript studied the property of a supervised representation learning, called DIET, and then propose an improved version of this method. Specifically, the authors show the equivalence between DIET and spectral contrastive loss proposed by HaoChen \& Ma under a linear case. In addition, the improvement is motivated by the insight derived from the model introduced in the setting of Section 5.1. Although it looks interesting when all the strong assumptions are true, it is unclear if the results presented in the manuscript can provide guidance in the practical settings.

**Strengths:**

The connection between DIET and spectral contrastive loss is interesting.

**Weaknesses:**

The analysis is built on some unrealistic assumptions. It is unclear whether the results and development in the manuscript can hold in practice.  It could be more meaningful to make assumptions more carefully.

**Questions:**

1) The review in the manuscript is clearly incomplete. Some important literature are missing. The author may want to have a more comprehensive literature review.

2) The analysis in the Section 4 is not surprising due to the linear setting. Can the analysis be extended to the nonlinear case? The author may read the recent development under the nonlinear setting in Wang 2023.

3)Assuming $W_H$ is an isometry seems a very strong assumption. Do the authors put a constraint for $W_H$ in the loss function to enforce such isometry? Otherwise, the authors may consider removing this assumption.

4)The form of training example introduced in Section 5 is unrealistic. When does this assumption hold (at least approximately) in piratical example? Why are there two features, one is low noise and the other is high noise?

5)Can we generalize the results in Theorem 5.1 to a more realistic setting?

6) The idea in memory-efficient DIET is straightforward.

---

> ### Author Response · Authors · 2024-11-21
>
> We thank the reviewer for their comments on our paper.  While we have detailed the contributions of our work in the general comment [(link)](https://openreview.net/forum?id=4NgxI6Z74n&noteId=Buw6JHWcGS), we would like to address the reviewer’s specific remarks in detail here.
>
> While the reviewer raises some concerns about the theoretical assumptions made in our work, we note that our assumptions are commonly used in the machine learning theory literature. Moreover, our empirical results in Section 6 confirm the validity of our results in practical scenarios. We hope to address each of the reviewer’s points in detail below:
>
> 1. While we attempted to provide a broad overview of existing work on contrastive self-supervised learning (Chen et al., 2020; He et al., 2020; Oord et al., 2018; Grill et al., 2020; Chen & He, 2021; Zbontar et al., 2021; Peng et al., 2022; Yang et al., 2022; Dwibedi et al., 2021) and the theory behind it (Wang & Isola, 2020; Graf et al., 2021; Arora et al., 2019; HaoChen et al., 2021; Lee et al., 2021; Tosh et al., 2021; Wen & Li, 2021; Ji et al., 2021; Saunshi et al., 2022; HaoChen & Ma, 2022; Xue et al., 2023; Xue et al., 2024; Balestriero, 2023; Murphy, 2022), it is possible that we missed some papers in the literature review. If the reviewer has any specific papers that they feel must be included, we would be happy to add those in the revised version.
>
> 2. Indeed, the analysis in Section 4 is surprising even in the linear case, given that the contrastive loss, which is based on the similarity between pairs of examples, and DIET, which is based on classifying examples based on pseudolabels, look vastly different. Additionally, to our knowledge, ours is the first fully rigorous equivalence between contrastive learning and supervised learning. While the exact equivalence does not hold for nonlinear models, our experimental results in Section 6 and Appendix D suggest that the equivalence approximately holds. Exploring this connection rigorously would be an interesting idea for future work. We were not able to identify which paper is Wang 2023, if the reviewer can provide a full reference we would happily provide a comparison.
>
> 3. We do not put an explicit constraint to enforce that $W_H$ is an isometry. While $W_H$ is unlikely to be an isometry in practice, [1] showed in theoretical and empirical examples that a linear projection head only performs feature rescaling, which is related to the phenomenon of neural collapse [2]. Thus our results hold up to rescaling even if we do not explicitly enforce that $W_H$ is an isometry. We have added this discussion in the revised manuscript.
>
> 4. Our data model in Section 5 is a variant of the sparse coding model, which has been widely used in previous work [1,3,4,5]. The use of a low noise feature and a high noise feature is to show that standard DIET learns mostly the low noise feature but DIET with normalization can learn both features. As a simple example, when identifying animals, say dogs versus birds, a low noise feature for birds could be the presence of wings (all birds have wings), while a high noise feature could be feather color, and the background would be noise. Besides, the ablations in Section 6 validate the effectiveness of normalization on more complicated real-world datasets.
>
> 5. The data model in Section 5 can be generalized in a few directions, for example, by having more features per class or having a variable number of features in each example. However, this makes the analysis more tedious without providing any extra insights, so we chose the simple setting for clarity.
>
> 6. The goal of our work is to show that our relatively simple method S-DIET can match the performance of CL and more complicated SSL methods. The fact that such a simple method can achieve state-of-the-art performance while being more memory-efficient is indeed surprising and we believe is worth sharing with the wider community. Meanwhile, more advanced modifications that further improve performance are left to future work.
>
> [1] Yihao Xue, Eric Gan, Jiayi Ni, Siddharth Joshi, and Baharan Mirzasoleiman. Investigating the benefits of projection head for representation learning. 2024.
>
> [2] Papyan, Vardan, X. Y. Han, and David L. Donoho. Prevalence of Neural Collapse during the Terminal Phase of Deep Learning Training. 2020.
>
> [3]: Zixin Wen and Yuanzhi Li. Toward understanding the feature learning process of self-supervised contrastive learning. 2021.
>
> [4]: Zou, D., Cao, Y., Li, Y., and Gu, Q. Understanding the generalization of adam in learning neural networks with proper regularization. 2021.
>
> [5]: Chen, Y., Huang, W., Zhou, K., Bian, Y., Han, B., & Cheng, J. Understanding and Improving Feature Learning for Out-of-Distribution Generalization. 2023.

---

> > ### Author Response · Authors · 2024-11-24
> >
> > We hope our rebuttal has addressed your concerns. As we are getting close to the end of the discussion session, we would like to reach out and see if you had a chance to read our rebuttal and if there is anything else we can further clarify? We are looking forward to your response.

---

> > ### Comment · Reviewer_umd8 · 2024-11-26
> > **Thanks for the response**
> >
> > Thanks for your response. The response cannot fully address my concerns. Specifically, a result held in a linear case does not necessarily have to be held in a more general case. Some discussions are needed to explain why the "insight" obtained from the linear model holds in a general setting. The data model in Section 5 is too far away from the practice. A more realistic setting can be considered. Addressing these comments may need substantive work.

---

### Official Review · Reviewer_nqvr · 2024-11-03

**Soundness:** 3
**Presentation:** 1
**Contribution:** 2
**Rating:** 5
**Confidence:** 4

**Summary:**

This paper presents S-DIET, a memory-efficient modification of DIET for self-supervised contrastive learning. It proves that DIET with a linear encoder and MSE loss is theoretically equivalent to spectral contrastive loss, and proposes feature normalization and projection head use to enhance performance. S-DIET significantly reduces DIET's memory requirements and achieves state-of-the-art performance without extensive hyperparameter tuning.

**Strengths:**

1. The paper provides comprehensive and rigorous theoretical proofs.
2. Addressing the high memory demand of DIET is a well-motivated objective with strong practical significance.
3. The experimental results presented in Table 5 demonstrate promising improvements.
4. The paper conducts a detailed and insightful ablation study.

**Weaknesses:**

1. The paper aims to address three issues: why DIET can perform comparably to CL, DIET's failure to learn all features, and its high memory demand. However, these issues are addressed in a fragmented manner without clear logical connections between them, making it difficult for readers to grasp the paper's central thesis.
2. The paper does not provide code or pseudocode, which hinders understanding of the proposed method and limits the ability to verify its effectiveness.
3. It is well-known that MSE is not typically used as a loss function for classification tasks. In the original DIET paper, each sample is treated as a separate class in a classification problem using cross-entropy loss. Why, then, is MSE employed as the loss function in Section 4? Does Theorem 4.5 hold if cross-entropy loss is used instead?
4. Due to the use of W1, it is unclear how Theorem 4.3 is related to the proposed method.
5. The theoretical analysis in the paper relies entirely on linear assumptions, while the proposed method (Equation 5) is based on empirical assumptions. These assumptions raise concerns about the rigor of this work.

**Questions:**

Please refer to the weakness.

---

> ### Author Response · Authors · 2024-11-20
> **Part 1**
>
> We appreciate the positive feedback from the reviewer on our rigorous theoretical results and the efficacy and significance of our proposed method S-DIET. We also thank the reviewer for providing detailed comments about our work. We hope to address each of the reviewer’s concerns below:
>
> 1. The main thesis of the paper is that S-DIET is theoretically as powerful as CL while providing practical advantages (less memory usage). CL has been extensively studied in the literature and several additions have been proposed that are essential to obtain optimal performance. To confirm S-DIET’s comparable performance to CL, we also studied multiple factors, i.e, normalization and projection head, that are common practice for representation learning with CL, and showed their effectiveness in boosting the performance of S-DIET. Nonetheless, we acknowledge the reviewer’s remarks and will connect each part to the central thesis in the updated version.
>
> 2. We thank the reviewer for the suggestion. We have included the following pseudocode for a single training step in the revised manuscript (page 27).
> ```
> """
> Uppercase variables stored on disk
> Lowercase variables stored in memory
>
> X: train data
> H: classifier head
> M: first moment for classifier head
> V: second moment for classifier head
>
> indices: indices for the current batch
> """
> def train_step(X, H, M, V, indices, model, criterion, optimizer):
>     # Load data, head weights, and head optimizer state into memory
>     inputs, head, optimizer_m, optimizer_v = X[indices], H[indices], M[indices], V[indices]
>     labels = [0, 1, ..., len(indices)-1]
>
>     # Forward and backward pass
>     outputs = head(model(inputs))
>     loss = criterion(outputs, labels)
>     optimizer.zero_grad()
>     loss.backward()
>     optimizer.step()
>     head, m, v = perform_multistep_adamw_head_update(head, m, v)
>
>     # Save head weights and head optimizer state
>     # Done asynchronously
>     H[indices], M[indices], V[indices] = head, m, v
>
>
> def perform_multistep_adamw_head_update(head, m, v):
>     g = head.grad
>
>     # first step
>     head = (1 - lr * weight_decay) * head
>     m = beta1 * m + (1 - beta1) * g
>     v = beta2 * v + (1 - beta2) * g * g
>     head = head - lr * m / (sqrt(v) + eps)
>
>     # all other steps
>     mu = beta1 / sqrt(beta2)
>     alpha1 = (1 - lr * weight_decay) ** (t - 1)
>     alpha2 = (alpha1 * lr * mu - lr * (mu ** t)) / (1 - lr * weight_decay - mu)
>
>     head = alpha1 * head - alpha2 * m / (sqrt(v) + eps)
>     m = (beta1 ** (t - 1)) * m
>     v = (beta2 ** (t - 1)) * v
>
> ```

---

> ### Author Response · Authors · 2024-11-20
> **Part 2**
>
> 3. The use of the spectral contrastive loss and MSE loss in place of the InfoNCE loss and cross-entropy loss is common practice in non-information-theoretic analysis due to the tractability of the MSE loss [1,2,3,4,5]. While the exact equivalence does not hold for the cross-entropy loss, our experimental results in Section 6 and Appendix D suggest that the equivalence approximately holds. Exploring this connection rigorously would be an interesting idea for future work. Nevertheless, we believe our existing result is valuable as it is, to our knowledge, the first precise, fully rigorous connection between contrastive learning and supervised learning.
>
> 4. Theorem 4.3 demonstrates an equivalence between the global minima of a linear model trained with the spectral contrastive loss and one trained with DIET (namely appending a classifier head $W_H$ and training $W$ and $W_H$ to minimize the MSE loss). This suggests that the simpler DIET methodology can replace complicated CL methods, so we developed S-DIET, a modification of DIET, as a simpler, memory-efficient alternative to CL.
>
> 5. We note that it is almost always the case in the ML community that theoretical analysis is performed in simpler settings that do not perfectly apply to practical scenarios. However, all our assumptions are commonly used in the literature, such as linear models with spectral contrastive loss or MSE loss in Section 4 [1,2,3,4,5], or the sparse coding data model in Section 5 [6,7,8,9]. Moreover, the gap between theoretical and practical scenarios does not affect the rigor of the results: our proof of the equivalence between the global minima of the spectral contrastive loss and DIET with MSE loss for linear models is fully rigorous, and our experimental results show the efficacy and memory efficiency of S-DIET in real-world settings regardless of any theoretical results. The theoretical and empirical results provide two different points of view to justify our claim that DIET can match the performance of more complicated SSL methods.
>
> [1]: Zhou, J., Li, X., Ding, T., You, C., Qu, Q., & Zhu, Z. On the Optimization Landscape of Neural Collapse under MSE Loss: Global Optimality with Unconstrained Features. 2022.
>
> [2]: HaoChen, J. Z., Wei, C., Gaidon, A., and Ma, T. Provable guarantees for self-supervised deep learning with spectral contrastive loss. 2021.
>
> [3]: Saunshi, N., Ash, J., Goel, S., Misra, D., Zhang, C., Arora, S., Kakade, S., and Krishnamurthy, A. Understanding contrastive learning requires incorporating inductive biases. 2022.
>
> [4]: HaoChen, J. Z. and Ma, T. A theoretical study of inductive biases in contrastive learning. 2022.
>
> [5]: Yihao Xue, Siddharth Joshi, Eric Gan, Pin-Yu Chen, and Baharan Mirzasoleiman. Which features are learnt by contrastive learning? on the role of simplicity bias in class collapse and feature suppression. 2023.
>
> [6] Yihao Xue, Eric Gan, Jiayi Ni, Siddharth Joshi, and Baharan Mirzasoleiman. Investigating the benefits of projection head for representation learning. 2024.
>
> [7]: Zixin Wen and Yuanzhi Li. Toward understanding the feature learning process of self-supervised contrastive learning. 2021.
>
> [8]: Zou, D., Cao, Y., Li, Y., and Gu, Q. Understanding the generalization of adam in learning neural networks with proper regularization. 2021.
>
> [9]: Chen, Y., Huang, W., Zhou, K., Bian, Y., Han, B., & Cheng, J. Understanding and Improving Feature Learning for Out-of-Distribution Generalization. 2023.

---

> > ### Author Response · Authors · 2024-11-24
> >
> > We hope our rebuttal has addressed your concerns. As we are getting close to the end of the discussion session, we would like to reach out and see if you had a chance to read our rebuttal and if there is anything else we can further clarify? We are looking forward to your response.

---

> > > ### Comment · Reviewer_nqvr · 2024-11-25
> > > **Thanks for the reply**
> > >
> > > Thank you for your response. While it provides some clarification, it does not fully address the concerns I raised. First, the updated pseudocode is somewhat helpful for understanding the methodology. Second, I acknowledge the authors' statement that the theoretical foundation is based on simplified assumptions. However, the revised version still fails to resolve the lack of logical coherence, which remains my primary concern. Addressing this issue, in my view, would require a major revision to the manuscript. As I noted in W1, the current outline attempts to address too many issues without focusing sufficiently on a single core problem.

---

### Official Review · Reviewer_T8fX · 2024-11-04

**Soundness:** 2
**Presentation:** 1
**Contribution:** 1
**Rating:** 3
**Confidence:** 3

**Summary:**

This paper studies DIET, which is a method that has rather few adoptions. This paper claims that DIET uses fewer memory than common contrastive learning approaches, and proved some theoretical results showing that DIET and spectral contrastive learning share the same solutions. Moreover, this paper proposes a new alternative S-DIET to further improve its performance.

**Strengths:**

1. This paper has theoretical claims that connects DIET to CL.
2. This paper has some empirical evidence that the proposed S-DIET can match the performance on benchmarks like CIFAR and ImageNet-100.

**Weaknesses:**

The method DIET studied in this paper has a fatal limitation, which is that the labels are essentially the sample index. As the dataset size increases, the classification head will need to increase linearly as well, which makes it impractical to use in large scale dataset training. Even though the proposed S-DIET does not require the classification head to be always loaded into the memory, it is still unnecessary to store such a large head, especially when one is training on millions of data.

**Questions:**

None

---

> ### Author Response · Authors · 2024-11-20
>
> We thank the reviewer for their response. We have provided some clarifications about the contributions of our work in the general comment [(link)](https://openreview.net/forum?id=4NgxI6Z74n&noteId=4X8V2lFySi), and will address the reviewer’s response in detail below.
>
> The reviewer correctly identifies that S-DIET maintains a large classifier head but the entire classifier head does not need to be held in memory. However, unlike DIET, storing the classifier head on disk is precisely why it is no longer a significant limitation when applying our method to large data. Specifically, the dataset itself has size $N \times d$ while the classifier only has size $N \times m$, where the embedding dimension $m$ is much smaller than the input dimension $d$. As an example, on ImageNet-1k, which has over 1.2 million images, using an embedding dimension of 2048 results in a classifier head with a size of approximately 10GB, whereas the ImageNet dataset itself is around 150GB. Thus, **storing the classifier head only adds negligible overhead to disk storage compared to storing the dataset. In exchange, our method requires half the GPU memory of existing CL methods**, as illustrated in Table 6. Lower GPU memory requirement has strong practical significance. For example, even with batch size 256 and a large A40 GPU, CL and other SSL methods nearly run out of memory when training a ResNet-50 on ImageNet data. Meanwhile, our S-DIET can use as little as half the memory as other SSL methods. Thus, we believe S-DIET is a promising alternative for SSL in memory-limited scenarios. We hope this clarifies the design choices made in the paper.

---

> > ### Author Response · Authors · 2024-11-24
> >
> > We hope our rebuttal has addressed your concerns. As we are getting close to the end of the discussion session, we would like to reach out and see if you had a chance to read our rebuttal and if there is anything else we can further clarify? We are looking forward to your response.

---

### Author Response · Authors · 2024-11-20

We would like to highlight the main motivation and contribution of our work, which is not fully captured in all the reviews. Contrastive Learning (CL) is one of the most popular and successful methods for representation learning. However, CL requires a large encoder and crucially relies on a large batch size to learn high-quality representation. Therefore, CL methods require a considerably large GPU memory. For example, the well-known SimCLR method trained their best model with CloudTPUs, using 128 cores and a batch size of 8192 to train encoders 4x larger than ResNet50 (Section 2.2 in [1])! In our work, we investigated an alternative “supervised” approach for representation learning (S-DIET), with considerably lower memory requirements. We theoretically proved the promise of S-DIET to learn high-quality representations, and empirically confirmed its effectiveness:
1. We proved a precise equivalence between solutions of S-DIET and spectralCL, which confirms that supervised approaches can learn high-quality representations. To our knowledge, this is the first fully rigorous connection between supervised and contrastive learning.
2. We showed that S-DIET can be implemented with much lower memory requirements, compared to CL.
3. To further boost the S-DIET performance, we proved in a simple theoretical example that normalizing embeddings enables S-DIET to learn noisier features in addition to the less-noisy ones.
4. We empirically showed that S-DIET matches the performance of state-of-the-art CL methods while using substantially less memory than all the other methods.
We believe our contributions show the promise of simpler and more efficient approaches for representation learning.

[1] Ting Chen, Simon Kornblith, Mohammad Norouzi, and Geoffrey Hinton. A simple framework for contrastive learning of visual representations, 2020.

---

### Note · Program_Chairs · 2024-10-23
**Submission Desk Rejected by Program Chairs**

margin violation

---

> ### Note · Program_Chairs · 2024-10-24
>
> **Comment:**
>
> The margin difference is not noticeable without a ruler (all the style difference of this paper is due to the usage of a legacy template), and the paper will fit into 10 pages with the correct template. After more cases emerges and based on more discusses, we decided to lean on the lenient side. Please proceed with reviewing of this paper.
>
> **Revert Desk Rejection Confirmation:**
>
> We approve the reversion of desk-rejected submission.

---

### Meta-Review · Area_Chair_x659 · 2024-12-17

**Metareview:**

The paper studies DIET, a method for self-supervised representation learning by labeling each sample as a distinct class and training a classifier as in standard classification networks. The main contributions include

- Theoretical Insight: The authors prove that DIET with a linear encoder trained with MSE loss is equivalent to the spectral contrastive loss, connecting supervised losses to contrastive learning frameworks.
- Feature Learning Limitation: DIET tends to prioritize learning less noisy features but may miss others. The authors demonstrate that feature normalization mitigates this issue, and a projection head further enhances performance.
- Memory Efficiency: S-DIET addresses DIET's memory inefficiency by storing the classification head on disk instead of memory. This significantly reduces GPU memory consumption.
- Empirical Results: S-DIET achieves state-of-the-art performance on benchmarks like CIFAR-10 and ImageNet-100 with lower memory usage and no extensive hyperparameter tuning.

Reviewers generally recognize the importance of the problem, appreciate the theoretical insights and the practicality demonstrated through experiments. The main concerns are the following:

- Unrealistic Assumptions: The theoretical analysis relies on linear model setting and specific data models, questioning the applicability to more general, practical cases.
- Fragmented Presentation: Reviewer nqvr criticized the paper for addressing multiple issues (theoretical equivalence, feature learning, and memory efficiency) in a "fragmented manner," which weakens the paper’s logical flow. Similar concern was raised by Reviewer cXwx who complained about fragmented conclusions and lack of motivation.

These concerns indicate that the paper requires improved writing to make the motivation and conclusion clearer to the audience of the conference and possibly more effort on the theoretical study.

In particular, while simplifying assumptions are necessary, the particular assumption of a linear network represents a significant departure from the practical use cases of contrastive learning.

With regard to the MSE loss, while the authors use Ref [1] (which provides a neural collapse analysis of MSE) to justify their choice, please be noted that neural collapse analysis has also been conducted for CE in earlier work.

Given these limitations, I’d place this interesting and potentially impactful study in a marginally below borderline area. I strongly encourage the authors to improve their manuscript addressing the clarity and theoretical assumption issues above for future submissions.

**Additional Comments On Reviewer Discussion:**

Concern: Theoretical assumptions are unrealistic and limited to the linear case.

Response: Authors defended the assumptions, stating they are standard in machine learning theory and empirically validated their findings in practical settings. However, Reviewer umd8 maintained that the results lack generalizability to non-linear models.

Concern: Fragmented presentation and unclear logical flow.

Response: Authors clarified the connections between different sections and emphasized the central motivation. Reviewer nqvr still felt the paper lacked logical coherence and required significant revision.

Concern: Motivation for adding projection head and feature normalization.

Response: Authors argued that these additions align with standard CL practices and improve performance. Reviewer cXwx felt this response did not adequately connect to the paper’s stated focus on memory efficiency.

Concern: Missing pseudocode for reproducibility.

Response: Authors included pseudocode for a single training step in the revised manuscript. Reviewer nqvr acknowledged this improvement.

---

### Decision · Program_Chairs · 2025-01-22

Reject